# Auditory cortex conveys non-topographic sound localization signals to visual cortex

Camille Mazo [1] ✉, Margarida Baeta[1] & Leopoldo Petreanu [1] ✉

Spatiotemporally congruent sensory stimuli are fused into a unified percept. The auditory cortex (AC) sends projections to the primary visual cortex (V1), which could provide signals for binding spatially corresponding audio-visual stimuli. However, whether AC inputs in V1 encode sound location remains unknown. Using two-photon axonal calcium imaging and a speaker array, we measured the auditory spatial information transmitted from AC to layer 1 of V1. AC conveys information about the location of ipsilateral and contralateral sound sources to V1. Sound location could be accurately decoded by sampling AC axons in V1, providing a substrate for making location-specific audiovisual associations. However, AC inputs were not retinotopically arranged in V1, and audio-visual modulations of V1 neurons did not depend on the spatial congruency of the sound and light stimuli. The non-topographic sound localization signals provided by AC might allow the association of specific audiovisual spatial patterns in V1 neurons.

Integrating information from various specialized sensory organs amplifies our capacity to comprehend and respond to external stimuli[1]. Multimodal sensory stimuli from a common source are congruent in space and time and are bound together into a unified cross-modal object[2]. Sounds can alter our visual perception and vice versa, as experienced in cross-modal illusions such as the ventriloquism effect[3], the McGurk effect[4], and the double flash illusion[5]. However, the neuronal mechanisms by which spatially congruent audiovisual stimuli are bound together remain poorly understood.

Visual space is directly mapped at the surface of the retina and information about the spatial location of sound sources, while not directly available in the sensory organs, can be computed using auditory cues[6]. It is well established that spatially congruent auditory and visual stimuli are integrated in the optic tectum of birds and superior colliculus (SC) of mammals[1]. These structures contain spatial auditory and visual maps, and bimodal neurons whose spatial receptive fields (RFs) are in register (barn owl[7], ferret[8], cat[6], guinea pig[9], and mouse[10]). The neural register of auditory and visual representations in these structures provides a substrate for binding spatially congruent auditory and visual stimuli. Consistent with this, auditory and visual stimuli originating from the same location result in larger neural responses in the SC than when they are distant from each other[11,12].

In addition to the SC, it is becoming increasingly evident that sensory areas of the neocortex, even the primary ones, also participate in multimodal sensory integration[13]. Across species, visual responses in the primary visual cortex (V1) are modulated by sounds[14–23]. Direct projections from the auditory cortex (AC) to V1 are also conserved across species[23–28] and are thought to mediate at least some of the audiovisual interactions observed in V1 neurons[15,17,21,29] (but see[30,31]). AC neurons show tuning to sound locations[32–40]. Thus, by sampling AC afferents, neurons in V1, like those of the SC, could potentially have access to both auditory and visual spatial information. While AC does not harbor a topographic map of space[33,41], through selective innervation, spatial information in AC inputs could become aligned with the retinotopic map in V1, as feedback projections from higher visual areas do[42]. Such an organization could constitute a neural substrate for a cortical representation of spatially congruent audiovisual objects. Consistent with this, the spatial profile of sound-evoked activity has been reported to be aligned with V1 neurons' RF in cats[43,44].

However, while V1-projecting neurons are known to selectively encode sound features that are less represented within AC[17], it remains unknown whether they relay information about the location of sound sources, as required for V1 neurons to bind spatially congruent bimodal stimuli by sampling these inputs.

[1]Champalimaud Neuroscience Programme, Champalimaud Foundation, Lisbon, Portugal. ✉e-mail: camille.mazo@gmail.com; leopoldo.petreanu@neuro.fchampalimaud.org

In this study, we measured the auditory spatial information in AC afferents in layer 1 of mouse V1 and compared it to the RF of their postsynaptic neurons. We found that many AC to V1 (AC→V1) inputs in layer 1 have spatially restricted RFs, encoding spatial information that V1 neurons can potentially use to accurately locate sound sources. However, the auditory RFs in AC inputs in V1 span ipsi- and contralateral hemifields and bear no relation with the visual RFs of their postsynaptic V1 neurons. Consistent with this, we observed that visual responses in V1 neurons are equally enhanced by sounds, regardless of the degree of spatial congruence between the auditory and visual stimuli.

## Results

### AC inputs convey auditory spatial information to V1

To gain spatial control over auditory and visual stimuli, we designed an array of loudspeakers and light-emitting diodes (LED) spaced over a spherical section. The array consisted of 39 positions, distributed in 13 columns (10° steps) along the azimuth and 3 rows (20° steps) in elevation (Fig. 1a). An LED and a loudspeaker were mounted at each position. We also designed custom electronics boards and software to route signals to specific individual LEDs and speakers in the array (Methods). Mice were head-fixed in the center such that their head was equidistant to all the positions in the array. Mice faced the 3rd column and 2nd row, allowing stimuli to be presented over azimuthal positions spanning from 20° at the hemifield ipsilateral to the recorded hemisphere to 100° of the contralateral one, and 0° and 20° up and down in elevation relative to the head position (Fig. 1b). Thus, the array allowed presenting visual and auditory stimuli in locations spanning most of the azimuthal extent of the visual hemifield represented in V1. We presented interleaved auditory white noise bursts (bandlimited, 2-20 kHz) and flashing white LED lights (Fig. 1c).

We measured the spatial specificity of sound-evoked activity in AC→V1 projections using two-photon recordings of axons (Fig. 1d). We injected an adeno-associated virus (AAV) encoding the genetically encoded calcium indicator GCaMP6s into two sites along the rostro-caudal extent of the left AC (AAV1-hSyn-GCaMP6s; Supplementary Fig. 1a). By using transgenic mice (Thy1-jRGECO1a)[45], we simultaneously expressed the red calcium indicator jRGECO1a in V1 neurons and measured their visual RFs using the LED array (Fig. 1g, i, k). As neighboring V1 neurons within a field of view have similar visual RFs[46], this allowed comparing the auditory spatial information in AC→V1 axons to the visual RF of their postsynaptic neurons, despite being unable to identify them. To determine to what extent auditory RFs were specific to AC→V1 axons, we compared them against V2L axons[42] by injecting GCaMP6s in V2L in another cohort of mice (Supplementary Fig. 1b).

We ensured the accuracy of the injections post hoc by registering coronal histological sections to the brain atlas. We verified that in all the analyzed animals the injection sites were confined to AC or V2L and that the thalamic projections from GCaMP6s expressing neurons mainly targeted the medial geniculate nucleus or the lateroposterior nucleus in AC- and V2L-injected mice, respectively (Supplementary Fig. 1a-c). AC and V2L projections in V1 mainly innervated layer (L)1 but also sent sparser projections throughout the cortical depth as previously described (Fig. 1e, f)[21,47,48]. When imaged in vivo, both AC and V2L projections formed a dense mesh in L1 with many *en passant* boutons (Fig. 1g).

We extracted fluorescence traces from individual AC or V2L boutons and measured their responses to sounds and flashes of light at different locations (Fig. 1h, j). While a significant fraction of AC boutons were responsive to auditory stimuli (fraction responsive, observed vs. time-shuffled data: $p = 0.002$, two-sided paired $t$-test, $n = 8$ mice), the fraction showing light-evoked responses were not significantly larger than the shuffled data ($p = 0.08$) (Fig. 2a, b). In a subset of experiments, we confirmed that AC boutons were frequency-tuned, as expected

from auditory responses in AC neurons (Supplementary Fig. 2a–d). Onset responses were more abundant than offset ones (Supplementary Fig. 3a, b) as previously observed in V1-projecting AC neurons[17]. In V2L boutons, a large fraction showed visually-evoked responses ($p = 0.02$, $n = 3$ mice) while a smaller subpopulation of non-visually responsive V2L boutons responded to auditory stimulation ($p = 0.05$; Fig. 1j and Fig. 2a, b).

Auditory responses in AC boutons were often modulated by sound location while those of V2L boutons tended to be spatially homogenous (Fig. 1h, j; Fig. 2c). Visual responses in V2L boutons were also modulated by LED location, as expected from their known visual RFs[42] (Fig. 1j; Fig. 2c). A larger fraction of the auditory responses in AC boutons was modulated by the azimuth compared to elevation of the sound source (Fig. 2c). This difference was not due to the larger angular span in azimuth of the speaker array as it was preserved when restricting the analyses to stimuli from 40° x 40° isotropic arrays of 9 speakers (20° step; $p = 0.027$, $n = 8$ mice, two-sided paired t-test; Methods).

In each recording session, we observed a diversity of spatial response profiles in auditory responses in AC boutons, varying from entire or hemifield responses to narrowly tuned responses for different locations (Fig. 1h). In contrast, responses in V2L boutons were less varied. Most auditory-responsive boutons similarly responded to auditory stimuli in all locations, and visually responsive boutons responded to restricted spatial locations (Fig. 1j). To quantify the extent to which bouton responses were modulated by space, we calculated the spatial modulation index (SMI) of each bouton (Methods). The SMI has a value of 1 when responses are confined to a single location and 0 when their amplitude is equal across all locations (see SMI for example boutons in Fig. 1h, j). The SMI was largest for visual responses in V2L boutons, as expected from the known spatial selectivity of visual cortex neurons (Fig. 2d). The auditory responses of AC→V1 boutons were, on average, more spatially selective than the auditory responses of V2L boutons but less selective than their visual responses (Fig. 2d).

We measured the population spatial modulation in AC and V2L inputs to V1 by averaging all the responsive boutons and calculating the SMI. The averaged visual response across V2L boutons was restricted to specific LEDs, as expected from the retinotopic specificity of this projection (Fig. 2e)[42]. On the contrary, average auditory responses in both AC and V2L inputs were widely spread across all the speaker locations (Fig. 2e). Accordingly, the SMI of the average response per imaged location was lower for auditory responses in both projection types when compared with the visual responses of V2L boutons (Fig. 2f).

The position eliciting the largest response varied across AC boutons and spanned the full range of the sampled azimuthal positions (Fig. 2g). As for visual responses in V2L boutons, sound-evoked responses in AC boutons decreased monotonically with increasing distance from the boutons' preferred location (Supplementary Fig. 2e, f).

The previous observations showed that AC→V1 inputs have spatially confined auditory RFs that are on average broader than the visual RFs of V2L→V1 inputs. We next assessed whether the spatial information conveyed by AC→V1 inputs is sufficient to decode the position of the stimulus on a single-trial basis using a naïve Bayesian decoder (Methods). Within each individual imaging session, we grouped highly correlated boutons into one functional unit as they are likely to belong to the same axon[49,50]. We subsequently refer to these functional units as axons. We measured the ability of the decoder to accurately locate sounds on a per-trial basis using all the responsive axons from one imaging session. The distribution of the likelihoods across stimulus locations obtained by the decoder displayed a spatial component, with decaying likelihood away from the actual stimulus position (Fig. 3a). We measured the distance

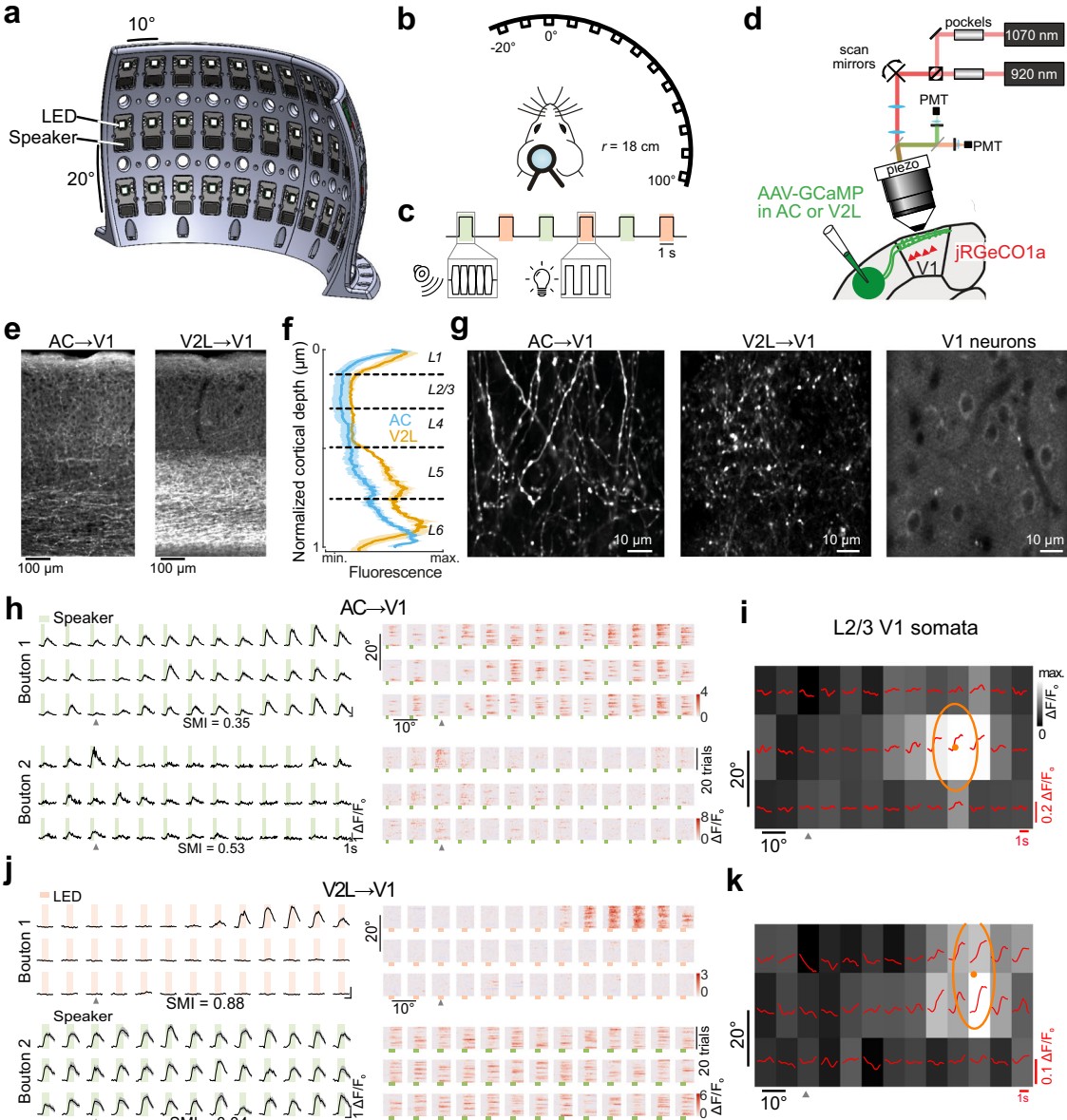

**Fig. 1 | Measuring auditory and visual receptive fields in AC and V2L inputs and how they compare with those of their target V1 neurons. a** Rendering of the LED and loudspeaker array. **b** Spatial relation between the head-fixed mice and the array. **c**, Stimulus timeline with examples of auditory (green) and visual stimuli (salmon). **d** Schematic of the dual-color, volumetric two-photon calcium imaging configuration. **e** Coronal histological section of AC (left) and V2L axons (right) in V1. **f** Normalized fluorescence intensity from AC (blue) and V2L axons (yellow) at different cortical depths. Mean (solid line) ± s.e.m. across mice (shaded area); $n = 7$ AC-injected mice and $n = 3$ V2L-injected mice. **g** Representative examples of a two-photon field of view in L1 of GCaMP6s-expressing AC or V2L axons (out of 143 and 15 sessions, respectively) and jRGECO1a-expressing somatas recorded in L2/3 beneath the AC/V2L axons in V1 (out of 158 sessions). **h** Left, Responses in example AC boutons from the same field of view to sounds from the different speakers in the array. Data is mean ± s.e.m. Right, Single trial responses of the same boutons. Green box, sound presentation (1 s). Arrowhead, midline position. SMI, spatial modulation index. **i** Population visual responses of the jRGECO1a-expressing somatas in L2/3 underneath the boutons in **h**. Average fluorescence signals (red traces) and response over the stimulus presentation window (grayscale color map) from stimuli at the corresponding location in the speaker and LED array. Ellipse, fitted RF; dot, RF center. **j** Visual (top) and auditory (bottom) responses in example V2L boutons in the same field of view in L1 of V1. Salmon, visual stimulus presentation (1s). Data is mean ± s.e.m. **k** Population visual responses of L2/3 somata underneath the V2L boutons in **j**. Source data are provided as a Source Data file.

between the decoded and the actual stimulus location (decoding error, Fig. 3a). Using auditory responses in AC axons, the decoding error was smaller than that of the trial-shuffled data (Fig. 3b). Decoding performance did not depend on the position of the speaker or the retinotopic position in V1 where the axons were recorded (Supplementary Fig. 4a–c). Speaker position could be better decoded from onset than offset responses (Supplementary Fig. 3c). Spatially specific responses were also evoked in AC boutons

using lower sound levels, and speaker location could be decoded from them (Supplementary Fig. 5a–d). In contrast, speaker position decoding error from auditory responses in V2L axons was not significantly different from the trial-shuffle control (Fig. 3b; Supplementary Fig. 4a, c). The location of visual stimuli, however, could be accurately estimated from V2L axons, as expected from their spatially confined visual RF (Fig. 3b; Supplementary Fig. 4a, c). We compared the decoder performance by comparing the decoding

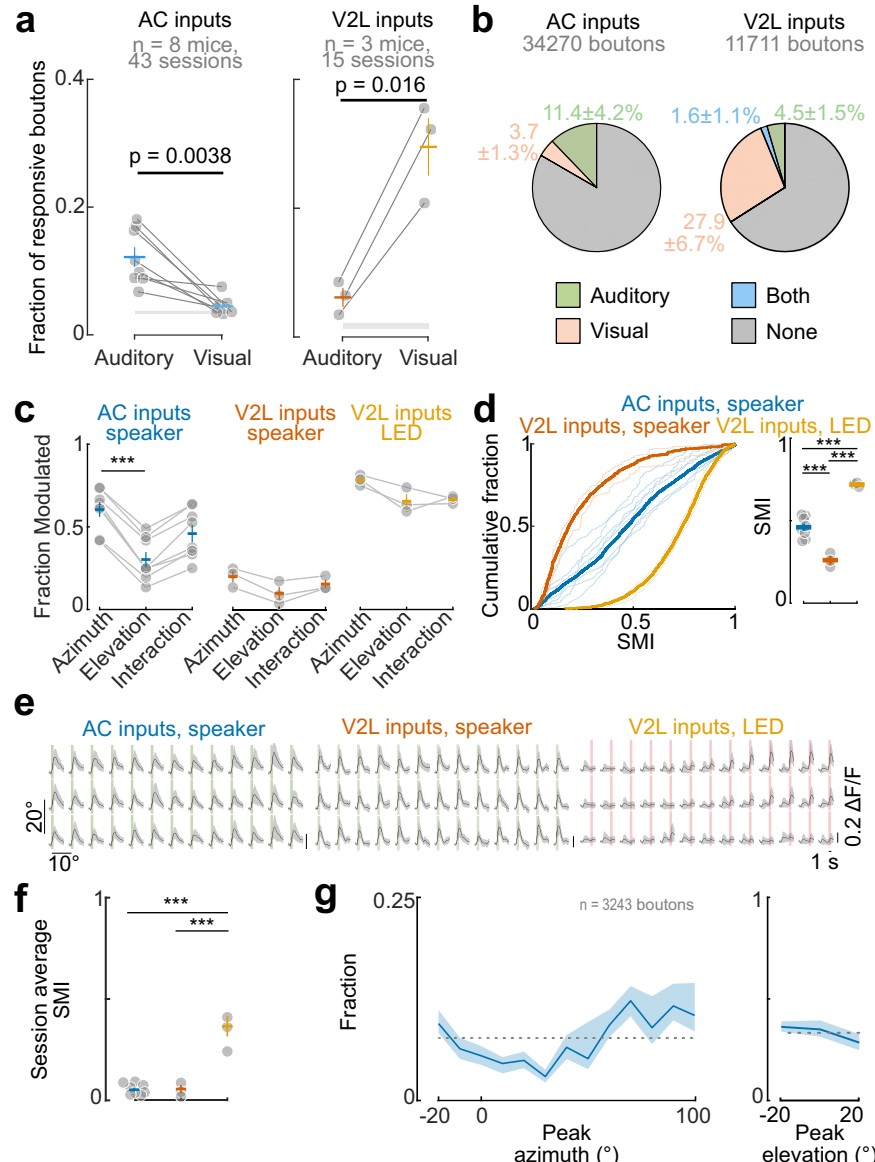

**Fig. 2 | Modality and spatial sensitivity in AC and V2L inputs to V1. a** Fraction of responsive boutons for sound and visual stimulus in AC and V2L inputs. Dots, individual mice; crosses, mean ± s.e.m. across mice; gray shaded area, mean fraction of responsive boutons from time-shuffled data ± s.e.m. across mice. Two-sided paired *t*-test. **b** Percentage of boutons responsive to auditory, visual or both stimuli in AC and V2L inputs. Percentages are mean ± s.d. across mice. Percentage of AC boutons responsive to both stimuli was <1%. **c** Fraction of boutons modulated by stimulus location. Dots, individual mice; crosses, mean ± s.e.m; $n = 8$ AC-injected mice and $n = 3$ V2L-injected mice. AC-injected mice: one-way repeated measure ANOVA, $p = 8 \times 10^{-4}$. Tukey's post-hoc test: azimuth vs. elevation, $p = 6 \times 10^{-4}$. **d** Left, Distribution of the spatial modulation index (SMI). 3-sample Anderson-Darling test, $p < 0.00001$. Thin lines, individual mice; thick lines, average per group. Right, Mean SMI. Dots, mice; cross, mean ± s.e.m. One-way ANOVA: $F(2,4) = 97.0$, $p = 2 \times 10^{-6}$. Tukey's post-hoc test: AC, speaker vs. V2L, speaker, $p = 5 \times 10^{-4}$; AC, speaker vs. V2L, LED, $p = 5 \times 10^{-5}$; V2L, speaker vs. LED, $p = 1 \times 10^{-6}$. **e** Example session averages across all responsive boutons to sound and LED flashes at different positions (same sessions as in Fig. 1h–k). Green shaded area, sound presentation; Salmon shaded area, visual stimulus presentation. Data is mean ± s.e.m. **f** SMI of the average across boutons. Each circle is the average per mouse. Crosses, mean ± s.e.m. One-way ANOVA: $F(2,11) = 46.5$, $p = 4.3 \times 10^{-6}$. Tukey's post-hoc test: AC, speaker vs. V2L, speaker, $p = 0.97$; AC, speaker vs. V2L, LED, $p = 4.6 \times 10^{-6}$; AC, speaker vs. V2L, LED, $p = 2.2 \times 10^{-5}$. **g** Distribution of the preferred azimuth (left) and elevation (right) across AC axon boutons. Line, mean; shading, 95% confidence interval. Dotted line denotes the uniform distribution. Source data are provided as a Source Data file.

error as a function of the number of axons sampled across projection types. Spatial accuracy using auditory responses in AC axons was greater compared to V2L axons but lower than when using visual responses in V2L axons (Fig. 3c).

We conclude that AC inputs relay signals encoding the location of sounds to V1 while inputs from V2L are largely devoid of such signals. The combined inputs from many AC boutons in L1 are sufficient for inferring the location of sounds but with lower accuracy than when locating visual stimuli using V2L boutons.

## Spatial auditory responses in AC inputs are independent of behavioral responses

Auditory modulations of neural activity in V1 neurons are thought to be driven, at least in part, by sound-evoked changes in internal state and uninstructed body movements[14,29,30]. However, the location-specific sound-evoked activity we observed in AC inputs is not consistent with the known features of behaviorally-driven cortical modulations. First, while behavioral modulation of neuronal activity is present across many brain structures[51–54], they are largely absent in the

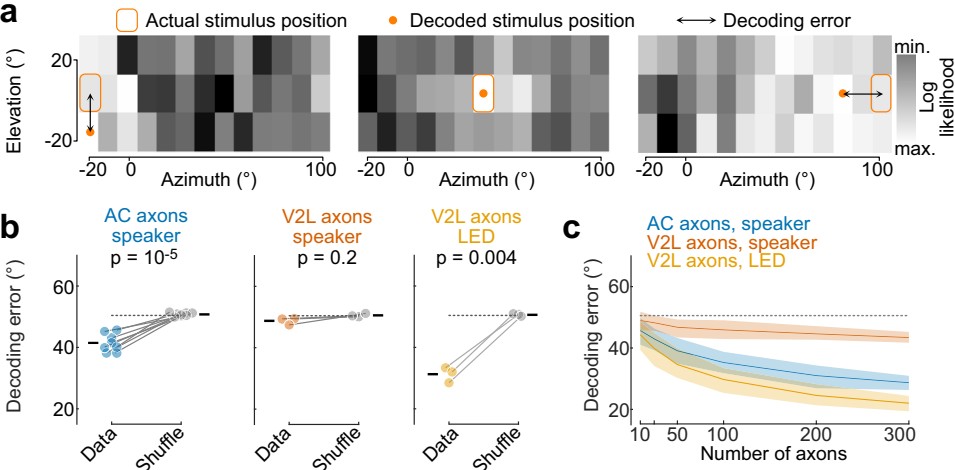

**Fig. 3 | Sound location can be decoded from AC inputs to V1. a** Log likelihood distribution across speaker positions for three example trials. **b** Distance between decoded and actual stimulus location. Colored dots, individual mice (average across imaging sessions); grey dots, shuffled data; ticks, mean across mice; dashed line, chance. Two-sided paired *t*-test: *n* = 8 AC-injected mice, 42 sessions; *n* = 3 V2L-injected mice, 15 sessions. **c** Distance between decoded and actual stimulus location as a function of number of axons. Solid line, mean; shading, 95% confidence interval; dashed line, chance. Source data are provided as a Source Data file.

AC[29]. Second, these modulations are low dimensional[30,51], i.e. they affect all neurons similarly, while the restricted spatial profile of the sound-evoked activity in AC→V1 inputs was diverse across boutons, even within the same session (Fig. 1h). To further confirm that spatial RFs in AC boutons were independent of uninstructed movements, we analyzed the behavioral responses to sound from the different locations by measuring pupil dilation in a subset of the experiments (5 AC- and the 3 V2L-injected mice). Sounds did induce pupil dilation (Supplementary Fig. 6a), but, on average, it was indifferent to speaker locations (Supplementary Fig. 6b). Consistently, speaker location could not be decoded from the pupil data (Supplementary Fig. 6c). Furthermore, in contrast to V2L→V1 boutons, sound-evoked responses in AC→V1 boutons were not correlated with pupil size (Supplementary Fig. 6d). We reached the same conclusion when analyzing facial movements, which have been shown to predict auditory-driven neuronal activity[30]. Sounds induced highly stereotypic facial movements (Supplementary Fig. 6e), independently of speaker location (one-way repeated measure ANOVA: azimuth, F(12,60) = 0.96, *p* = 0.50; elevation, F(2,10) = 0.69, *p* = 0.52; *n* = 6 mice), and these movements could not be used to decode speaker positions (Supplementary Fig. 6f). Finally, we also isolated imaging sessions in which sound did not induce any noticeable facial movements (Supplementary Fig. 6g). AC bouton responses in these sessions remained modulated by speaker positions (Supplementary Fig. 6h, i), and sound location could still be decoded from AC axons in V1 (Supplementary Fig. 6j). We conclude that, while the sound-evoked responses in V2L boutons are consistent with behavioral modulations, the spatial pattern of the sound-evoked responses in AC→V1 boutons is largely independent of them.

**Auditory spatial information in AC→V1 inputs is not topographic**
We then investigated whether the auditory spatial RFs of AC→V1 inputs were topographically organized with respect to the V1 retinotopic map. We measured the azimuthal position of the speaker that reliably elicited the strongest response for each AC bouton (best azimuth) and calculated the median position across boutons per session. We then compared the average best azimuth of the AC boutons with the azimuth center of the visual RF of the population of jRGeCO1a-expressing L2/3 V1 somatas below them (Fig. 1i, k and Fig. 4a).

The auditory RFs of AC boutons were not topographically organized in V1 as the average best azimuth bore no relation with the azimuth RF center of the V1 neurons below them (Fig. 4b). A similar

result was obtained when we first averaged responses across boutons to compute the population best azimuth (Supplementary Fig. 7) and when we measured the best azimuth using lower sound levels (Supplementary Fig. 5e). In contrast, the visual RFs of the V2L boutons were topographically organized and matched with those of the V1 neurons they target, as previously described (Fig. 4b, Supplementary Fig. 7)[42].

We checked if the location of sounds could be decoded differently from AC→V1 inputs when they correspond to the retinotopically-matched location in V1, despite the lack of retinotopic organization in the projection. We found that the decoder's performance was similar regardless of the speaker's location relative to the V1 azimuthal position where AC inputs were recorded (Fig. 4c), indicating no functional selective mapping of auditory space to visual space in V1. In contrast, decoding performance increased with increased proximity to V1 RF center in visual responses in V2L→V1 axons, consistent with the retinotopic alignment of these inputs (Fig. 4c).

**The absence of topographic organization in the auditory spatial responses of AC→V1 inputs is not due to a lack of high-frequency cues**
Recent work has shown that high-frequency cues (>20 kHz) are necessary for the topographic alignment of the auditory and visual RFs in azimuth in the SC[10]. As a consequence, C57BL/6 mice did not show a topographic auditory and visual alignment in the SC due to an early onset high-frequency hearing deficit[55]. Because Thy1-jRGECO1a mice were on a C57BL/6 background[45], they might be susceptible to high-frequency hearing deficits as well. Therefore, we repeated our experiments using broadband white noise that includes high frequencies (2–80 kHz) in CBA mice, as this strain does not suffer age-related high-frequency hearing loss[56] and contains more AC neurons representing high-frequencies[57]. We also presented these broadband stimuli to another group of Thy1-jRGECO1a mice.

We used AAV injections to transfect AC→V1 inputs with GCaMP8m[58] and V1 with jRGECO1a (in CBA mice). As before, AC→V1 boutons in both groups showed a rich variety of location-specific sound-evoked responses. Consistent with the role of high-frequencies in sound localization, in both CBA and Thy1-jRGECO1a mice, the speaker position decoding error was lower than that of the trial-shuffle data (Supplementary Fig. 8a) and more precise than when using bandlimited sounds in Thy1-jRGECO1a mice (Supplementary Fig. 8b). However, as before, we did not observe a correlation between the

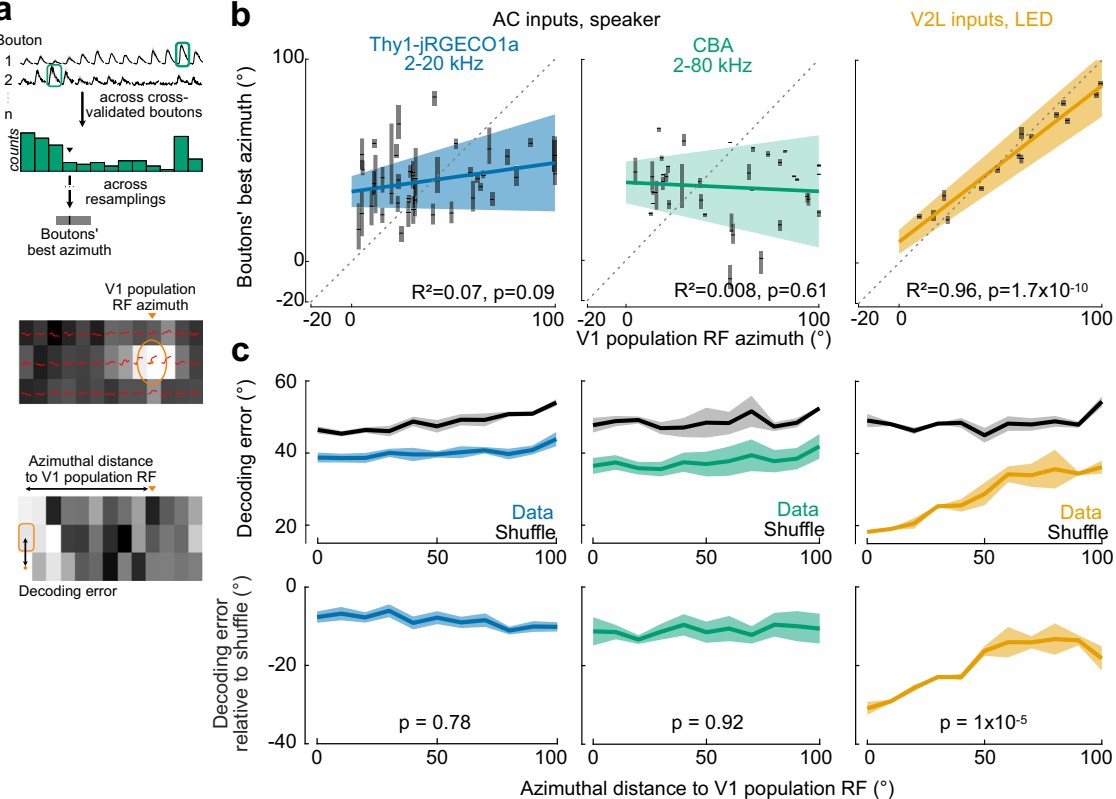

**Fig. 4 | AC inputs in V1 are not retinotopically matched. a** Top, Example bouton azimuthal responses, averaged across elevation. Same boutons as in Fig. 1h. Green boxes denote best azimuth. Median of the distribution (arrowhead) is used to estimate the population best azimuth. This is repeated over 100 iterations of resampling to estimate the average and 95% confidence interval (black bar and gray shaded area,). Middle, V1 somata population RF azimuth. Bottom, Decoding error in an example trial. The distance between the stimulus position and V1 population RF azimuth (top arrow) is represented on the abscissa in **c**. **b** Mean best azimuth across boutons as a function of the population RF center of V1 neurons for each imaging session. Left, AC boutons, Thy1-jRGECO1a mice, 2–20 kHz auditory white noise; $n = 41$ imaging sessions, 8 mice. Middle, AC boutons, CBA mice, 2–80 kHz auditory white noise; $n = 37$ imaging sessions, 6 mice. Right, V2L boutons, LED; $n = 15$ imaging sessions, 3 mice. Ticks, median; grey shading, 95% confidence interval.

Colored lines and values correspond to the linear regression of the mean values; colored shading, 95% confidence interval. Dashed lines, identity lines. **c** Top, stimulus location decoding error as a function of the distance between stimulus location and V1 population RF center. Left, AC axons of Thy1-jRGECO1a mice, 2–20 kHz auditory white noise. Middle, AC axons of CBA mice, 2–80 kHz auditory white noise. Right, V2L axons, LED. Two-way repeated measure ANOVA, interaction: AC axons 2-20 kHz white noise: $F_{(9,126)} = 0.86$, $p = 0.56$; $n = 8$ mice, 38 imaging sessions; AC axons 2–80 kHz white noise: $F_{(9,72)} = 0.46$, $p = 0.89$; $n = 5$ mice, 24 imaging sessions; V2L axons LED: $F_{(9,36)} = 7.69$, $p = 10^{-6}$; $n = 3$ mice, 12 imaging sessions. Bottom, shuffle-subtracted decoding error. One-way repeated measure ANOVA. Data is mean ± s.e.m. across mice. Source data are provided as a Source Data file.

auditory RF azimuthal centers and the visual population RF azimuthal center from V1 L2/3 somatas (Fig. 4b, Supplementary Fig. 8c). Similarly, decoding accuracy did not depend on the retinotopic position in V1 of the AC→V1 inputs (Fig. 4c). Thus, while the presence of high-frequency cues enhances the location-specific information available in AC→V1 inputs, the lack of a topographic organization we observed is not due to a deficiency of these cues in our measurements.

**Spatially specific sound responses in AC→V1 inputs are not due to differences across speakers**

We wondered if, despite having calibrated the frequency responses across speakers, the spatially specific sound responses of AC axons could still reflect unnoticed variations in the sounds produced by the different speakers in the array. To ensure that this was not the case, we mounted a single loudspeaker on a rotating arm and presented sounds at different azimuthal positions (Supplementary Fig. 9a). We confirmed that the spatially specific sound responses of AC axons obtained with the speaker array were similar to those measured using the rotating speaker, as assessed by their SMI and the performance in decoding azimuthal position (Supplementary Fig. 9b, c). Furthermore, like when using the speaker array for stimulation, the spatial auditory

responses in AC→V1 inputs lacked topographical organization when evaluated using a rotating speaker (Supplementary Fig. 9d).

**AC→V1 inputs represent the location of sounds in both the ipsi- and contra-lateral hemifields**

Many AC boutons in V1 exhibited peak responses when sound was presented in one of the two ipsilateral locations in our speaker array (-20° and -10° in azimuth in Fig. 2g). To measure to what extent AC inputs represent spatial locations outside the contralateral visual field of V1, we presented sounds over 180° around the head, spanning 90° ipsi- and contra-laterally using the loudspeaker mounted on a rotating arm (Fig. 5a). Consistent with the measurements using the speaker array, many AC→V1 axons were spatially tuned, and neighboring boutons showed different spatial RFs that, in many cases, peaked in different hemifields (Fig. 5b). Boutons tuned to high (>30°) contralateral and ipsilateral locations were more abundant than those tuned to frontal locations (Fig. 5c). Consequently, the locations of sounds originating in lateral locations in both hemifields could be decoded with similar errors, but those in front of the animals resulted in larger decoding errors (Fig. 5d). Thus, many AC→V1 inputs relay localization signals about sounds originating in regions outside the visual hemifield

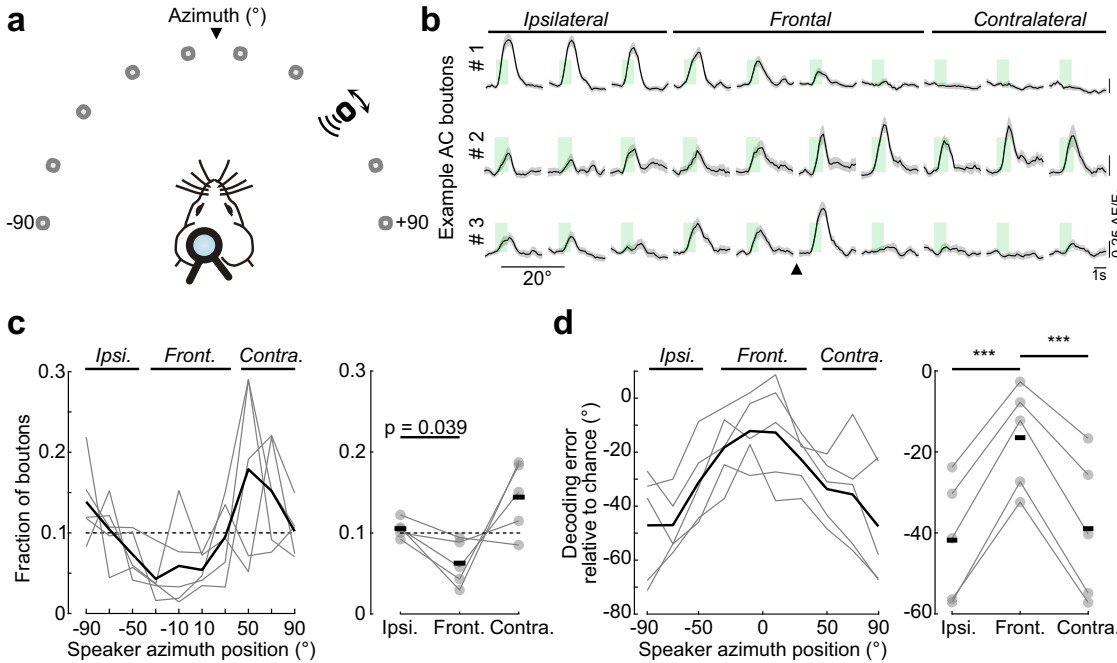

**Fig. 5 | AC inputs to V1 are tuned to sounds located in both ipsi- and contra-lateral hemifields. a** A single loudspeaker (black rectangle, broadband white noise) was moved between −90° to +90° in azimuth, 20° spacing (0° elevation; gray positions). Arrowhead, midline. **b** Example AC bouton responses to sounds played across the azimuthal positions. Solid lines, mean response; shaded area, s.e.m. Arrowhead, midline. **c** Left, distribution of best azimuth across boutons. Gray lines, individual mice; black line, mean across mice. 1344 cross-validated boutons from $n = 5$ mice. Dashed line, uniform distribution. One-way repeated measure ANOVA, $p = 0.01$. Right, fraction of ipsilateral-, frontal- or contralateral-preferring boutons, normalized to the number of positions per group. Circle and gray lines, individual mice; black tick, average; dashed line, homogeneous distribution. One-way repeated measure ANOVA ($p = 0.027$), followed by Tukey's post-hoc test, $n = 5$ mice. **d** Sound location decoding error, subtracted by chance level for each speaker position (left) and averaged across ipsilateral, frontal or contralateral positions (right). One-way repeated measure ANOVA ($p = 3.2 \times 10^{-6}$), followed by Tukey's post-hoc test, $n = 5$ mice. Ipsilateral- vs. frontal-preferring: $p = 2.9 \times 10^{-4}$; Contralateral- vs. frontal-preferring: $p = 2.8 \times 10^{-3}$. Source data are provided as a Source Data file.

represented in V1. Furthermore, sound localization signals about lateral locations are more prevalent than those encoding locations in front of the mouse.

## Audio-visual interactions in V1 neurons do not depend on the spatial coherence of the stimuli

The AC has been shown to be a source of sound-evoked modulations in V1 neurons[15,17,29]. We thus checked if V1 responses are influenced by the spatial congruence of auditory and visual stimuli, even though AC→V1 inputs were not retinotopically matched with V1. We recorded audio-visual responses from V1 L2/3 neurons using two-photon calcium imaging in mice constitutively expressing GCaMP6f in pyramidal neurons (Fig. 6a).

We first mapped the population visual RF of the neurons within the two-photon field of view using the LED array. We then presented light flashes within the RF either alone (V), or concurrently with bandlimited white noise sound bursts (AV) from a speaker at the same spatial location (0°) or ± 20° and ± 40° away in azimuth. We additionally played auditory stimuli alone at the same positions (A; Fig. 6a). As multisensory modulations depend on the intensity of the stimuli, we tested different combinations of LED brightness and speaker loudness. Auditory or visual stimulation alone evoked increasing responses with increasing stimulus intensity (Supplementary Fig. 10a, b). Yet, auditory responses were not significantly different depending on the speaker distance to the neurons' RF (Fig. 6b, c). Visual responses in L2/3 V1 neurons were on average larger in the presence of sound (Fig. 6e; two-sided paired t-test, $p = 0.0012$, $n = 8$ mice)[14,17,29]. However, the magnitude of the AV enhancement did not depend on the distance of the auditory and visual stimuli (Three-way repeated measure ANOVA; effect of position: $F(28,112) = 0.61$, $p = 0.66$; $n = 8$ mice; Fig. 6f), or their brightness or loudness (effect of

brightness, $F(14,112) = 0.23$, $p = 0.71$; effect of loudness, $F(14,112) = 1.37$, $p = 0.29$; $n = 8$ mice; Supplementary Fig. 10c, d). The magnitude of the A-only evoked responses or AV modulations did not depend on speaker position either when considering absolute (Fig. 6) or relative modulations (Three-way repeated measure ANOVA, effect of position on AV/V: $F(28,112) = 0.21$, $p = 0.93$; $n = 8$ mice). There were also no significant interactions between speaker position with loudness and brightness on the AV modulations or A-only responses when considering either of the two modulation metrics ($p > 0.05$). Consistent with these observations, the speaker location could not be decoded from either A-only or AV responses of V1 neurons (Fig. 6d, g).

In a subset of experiments, we checked if spatially matched and non-matched AV responses in V1 would be different in cases where the two stimuli were further separated. As before, AV responses were indistinguishable when the speaker was separated from the LED by 80° in azimuth and they were spatially matched (two-sided paired t-test: $p = 0.44$, $n = 5$ mice).

Sound elicited unstructured movements in a loudness-dependent manner. Loud sounds evoked reliable movements while quieter ones did not (Supplementary Fig. 10e, f, h). In contrast to this, the magnitude of the AV modulation was similar across loudness levels (Supplementary Fig. 10d). V1 neurons showed AV modulations and A-only responses at the lowest sound intensity even though sound-evoked movements were undetectable. Auditory responses and AV modulation remained independent of the distance between the speaker position and the neuron's RF or visual stimulus, respectively, when using quiet sounds (Supplementary Fig. 10g, i). This suggests that some components of the auditory-evoked activity and the sound modulations of the visual responses in V1 are motor-independent, as suggested previously[14,29]. Whether these motor-independent signals depend on AC→V1 inputs remains to be determined. While we cannot

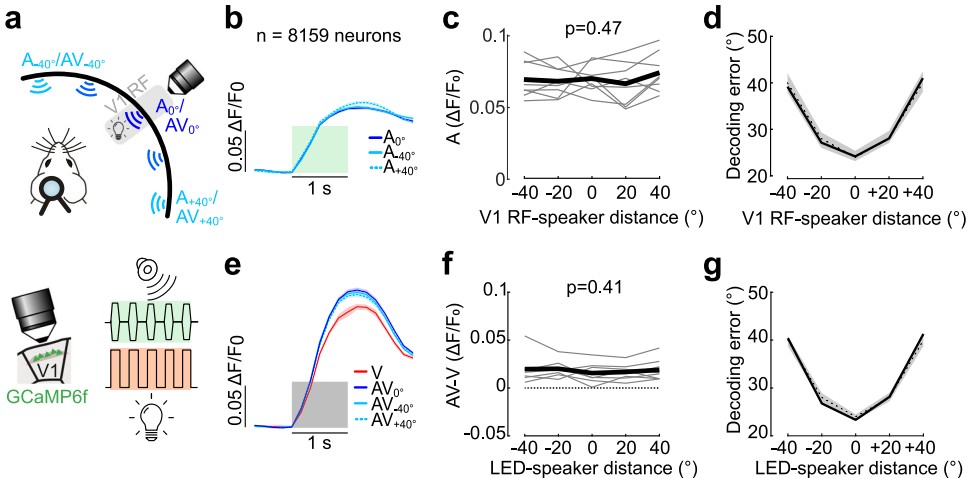

**Fig. 6 | Audiovisual interactions in V1 do not depend on the spatial coherence between the two stimuli. a** Imaging of GCaMP6f-expressing L2/3 neurons in V1 during audiovisual experiments. An LED located in the center of the population RF (gray shaded area; azimuth RF center range: 40-60°) was flashed while auditory stimuli were presented at one of the 5 azimuthal locations. 0° denotes matched speaker and LED position, negative and positive values correspond to speakers in lower and higher azimuthal positions, respectively. **b** Average auditory only (A) responses for 3 example speaker positions across all the recorded neurons (averaged across loudness levels). Data are mean across neurons ± s.e.m. Green shaded area, stimulation window (1 s). **c** Mean population response to A-only stimuli as a function of the distance between the speaker and the population RF of V1 neurons (averaged across loudness levels, one-way repeated measure ANOVA, *n* = 8 mice). Gray lines, average across neurons per mice; black line, average across mice. **d** Mean sound location decoding error (solid line) was within that of the shuffle data (shaded area, 95% confidence interval) for all distances between V1 neuron population RF and speaker positions. Dotted line, chance level. **e–g** Same as in **b–d** for audiovisual (AV) responses (averaged across brightness and loudness levels). V, visual trials (averaged across brightness levels). One-way repeated measure ANOVA, *n* = 8 mice. Source data are provided as a Source Data file.

completely dismiss the possibility that certain components of the signals may contain information regarding the spatial congruency of audio-visual stimuli or the positioning of sounds relative to the receptive fields of V1 neurons, these signals do not exert a predominant influence on V1 responses in mice when they are passively exposed to sounds.

## Discussion
We measured the relay of auditory and visual information from the AC to V1. Our findings show that while many AC inputs had broad spatial sound-evoked responses, others responded to sounds at specific ipsi- or contralateral locations. Through decoding analysis, we demonstrated that by sampling enough AC inputs in L1, V1 neurons could accurately estimate the location of sounds. Unlike visual RFs from other visual areas, the auditory RFs of AC inputs to V1 were nontopographic and did not bear any relation to the retinotopic map in V1. Consistent with the lack of topographic projection, we demonstrated that modulations of visually evoked responses in V1 did not depend on the spatial relationship of auditory and visual stimuli.

### AC relays information about the location of sounds to V1
We found auditory responses were much more frequent than visual ones in AC projections. Thus, in passively stimulated animals, unlike in mice that learned an audio-visual association[47], AC inputs to V1 are dominated by auditory responses over visual ones. These responses were independent of any uninstructed body movements and were consistent with somatic recording from V1-projecting AC neurons[17], confirming that auditory responses are present in AC inputs to L1 of V1.

As in the AC as a whole[33,35,36,41,59–61], AC afferents were spatially selective to varying degrees to sounds in specific ipsi- or contralateral locations. Given their broader spatial RFs, single AC boutons have limited information about the location of sounds. As a population, however, they have the capability to provide an accurate estimate of the location of sounds to postsynaptic V1 neurons. Thus, AC inputs in V1 provide a rich neural substrate from binding spatially and temporarily congruent AV stimuli.

On the contrary, inputs from V2L contained less information about the location of sounds than AC inputs, as expected from a visual area. However, sound location could be decoded slightly above chance when integrating V2L axons from multiple sessions and animals (Fig. 3c). The origin of the weak sound localization signals in V2L axons is unclear. Since sound-evoked responses in these axons exhibit stronger correlations with behavior than those originating in the auditory cortex (Supplementary Fig. 6d), as also observed in somatic recordings[29], these signals may indicate modulations linked to location-specific uninstructed movements. Nevertheless, it's important to note that sound location could not be decoded from facial movements and pupil dilations, which does not support this hypothesis. Hence, given the proximity of the two cortical areas[62], the weak sound localization signals in V2L inputs in V1 are likely inherited from sound localization signals originating in from the AC.

### Spatial information in AC inputs to V1 is not topographic
While many AC→V1 inputs had spatially confined auditory RFs, these differed from the visual RF of the V1 neurons they innervated in several aspects: 1) AC inputs to V1 were less spatially specific than the visual RFs of V1 neurons (Fig. 3 and Fig. 4), consistent with somatic recordings from AC neurons[33,63]. 2) In addition, the total span of the auditory RFs in AC inputs to V1 was larger than the visual field coverage of V1. That is, while within each hemisphere, mouse V1 mainly represents the contralateral visual space and only a small portion of ipsilateral frontally facing space, a large fraction of AC inputs was selective for ipsilateral sounds originating outside this range (Fig. 5). Thus, AC afferents in V1 relay information about sounds located outside the field of view of the targeted area. 3) We did not find any evidence of a topographic organization in the auditory spatial information relayed by AC to V1. This was true even in experiments with sound stimulus covering the full auditory spectrum (2-80 kHz white noise) and in mice that do not suffer from age-related high-frequency hearing loss[56,64]. While AC is tonotopically organized, it harbors no map of auditory space across species[41], including mice[33]. Our observations show that even when innervating an area with an accurate spatial map such as V1, AC inputs

remain non-topographically organized. Consistent with this absence of retinotopic organization, sound-induced modulations of visual responses did not depend on the spatial distance between the two stimuli (Fig. 6). In the superior and inferior colliculi, on the other hand, auditory and visual maps align, and responses are specifically enhanced by spatially congruent bimodal stimuli[10–12,65,66]. Thus, even though neurons in the two cortical areas share a common representational axis, i.e., they both represent the location of the stimuli, the connections between them do not follow a like-to-like connectivity. This contrasts with projections from higher-order visual and frontal areas to V1 that are topographically organized to match the retinotopy of V1, e.g. the visual RF of the afferent inputs are on average matched with those of their targeted V1 neurons (Fig. 4b)[25,42,67,68]. This difference in functional connectivity might be due to several features. On one hand, unlike AC inputs, higher-order visual and frontal areas convey spatial information from the same modality to V1. Sound location is encoded in head-centered coordinates, while light source location is encoded in eye-centered coordinates. Our experiments were performed in head-fixed mice, yet pinna position affects the relation of spectral cues to sound location. Thus, variations in both ear and eye positions might preclude the precise alignment of visual and auditory spatial maps. Yet, in the SC of mice, auditory and visual maps are aligned[10], showing that audiovisual spatial correspondence is possible despite the different coordinate frames. On the other hand, AC inputs to V1 are weakly reciprocated, unlike those from higher-order visual and frontal areas, as AC inputs to V1 are considerably stronger than those in the opposite direction in mice and other species[28,69]. Thus, retinotopically matched interactions across cortical projections might require recurrent interactions that are present in many cortical feedback inputs to V1[48,70,71] but are absent in the AC→V1 projection.

As our measurements were obtained from inputs terminating in L1, we cannot rule out the possibility that afferents terminating in deep layers (Fig. 1e, f) might harbor different sound localization signals than those terminating in L1 or that they would be topographically organized.

### Implications for the functional role of AC afferents in V1

What could the role of the non-topographic sound location encoding signals relayed from AC to V1 be? The organization of direct AC→V1 cortico-cortical inputs and the independence of AV modulations on the spatial coherence of the two stimuli differs from the spatial register found in the SC and the inferior colliculus[1,10,65]. These differences suggest that audiovisual interactions mediated by direct cortico-cortical connections might play a fundamentally different role than in midbrain structures.

Direct projections from AC to V1 are present in many species, suggesting a conserved function. Bimodal stimuli that are congruent in space and time result in perceptual and reaction time improvements in various tasks[1]. However, the role played by projections directly linking sensory areas of the neocortex remains unclear[13]. Given that auditory cortical responses are faster than visual ones, one possibility is that direct AC inputs sensitize V1 neurons with RFs overlapping with salient audiovisual objects, drawing attention to them and facilitating their discrimination[72,73]. Consistent with this, visual stimuli are perceptually more salient when they are in spatial register with the auditory ones[74–76]. However, such a mechanism would require aligned retinotopically matched AC afferents in V1. In addition, this hypothesis leaves inputs from AC neurons with RFs located outside the span of the innervated V1, such as from laterally located ipsilateral positions, without a function. One possibility is that V1 neurons integrate sound localization signals from AC inputs in L1 with upcoming motor commands for oriented body and head movements relayed from other sources. By integrating motor commands, V1 neurons could re-reference auditory localization signals in AC axons to retinotopic coordinates and become sensitized to the visual stimuli the AC inputs

anticipate[72]. Such a mechanism would require spatially specific sound signals in AC inputs, but these do not have to be necessarily retinotopically matched, as observed here.

In spatial tasks, the visual sense often prevails as it harbors more precise spatial information[77]. Hence, the lack of topographic organization in AC→V1 inputs might reflect that, while integrating sound information with visual inputs, V1 is not engaged in the detection of low-level features, such as the location of sounds. Consistent with this, inactivating V1 did not affect the performance of mice in a sound localization task[78]. Instead, it might utilize the diverse and distributed spatial information provided by AC→V1 inputs for higher-level analyses of auditory space, such as determining the relative positions of various sound sources for integration with visual information.

We cannot dismiss the possibility that topographic AC→V1 projections might develop in mice exposed to a more diverse multisensory environment than what is typically found in standard laboratory home cages. Additionally, retinotopic-specific projections could potentially emerge through training in tasks that demand the integration of auditory and visual cues for object localization. Future studies addressing how sound location-dependent visual modulations depend on experience will provide insights into how V1 neurons integrate the rich auditory spatial signals available in L1.

## Methods

### Animals

Thy1-jRGECO1a (Tg(Thy1-jRGECO1a)GP8.20Dkim/J, MGI:J:268005, JAX stock #030525), CBA/CaCrl (Charles River Laboratories, stock #609) and F1 offspring from the Slc17a7-IRES2-Cre (or Vglut1-IRES2-Cre-D, JAX stock #023527) x Ai148D (Ai148(TIT2L-GC6f-ICL-tTA2)-D, JAX stock #030328) crossing male and female mice were used in this study. Thy1-jRGECO1a mice were originally created on a C57BL/6J genetic background[45] and the line was maintained in the same background by backcrossing heterozygous mice. C57BL/6J were bred in-house. Mice were group housed, maintained under a 12:12 h regular light:dark cycle, at 23 C° with an humidity of 45–65%, and were provided food and water *ad libidum*. All animal procedures were reviewed by the Champalimaud Centre for the Unknown Ethics Committee guidelines and performed in accordance with the Portuguese Direção Geral de Veterinária.

### Surgeries

Surgeries were performed on young adult mice (8-9 weeks old, males and females) under isoflurane anesthesia (1.5%). Bupivacaine (0.05%; injected under the scalp) and buprenorphine (100 μl, 0.1 mg/kg, subcutaneously) provided local and general analgesia, dexamethasone (100 μl, 2 mg/kg, subcutaneously) and sodic cefovecin (100 μl/10 g, 6 mg/kg, subcutaneously) were used to minimize inflammation and infection, respectively. Eyes were protected and kept moist using ophthalmic ointment (Vitaminoftalmina A, Labesfal). Glass injection pipettes (Drummond) were beveled at 45° with a 13–18 μm inner diameter opening and backfilled with mineral oil. A fitted plunger controlled by a hydraulic manipulator (Narashige, MO10) was inserted into the pipette and used to load and inject the viral solution. The skull was thinned above the auditory cortex and the glass injection pipette was lowered through the remaining bone down to the auditory cortex. We injected 50-100 nL of AAV (AAV2/1-Syn-jGCaMP8m-WPRE, Addgene #162375 or AAV2/1-Syn-GCaMP6s-WPRE-SV40, Addgene #100843) either in the AC in two locations (Coordinates, posterior and lateral from Bregma, depth from brain surface: 2.5 mm, 4.3 mm, 0.8 and 1.2 mm depth and 3.0 mm, 4.7 mm and 0.5 mm depth), or in V2L (3.55 mm, 3.65 mm, 0.25 and 0.55 mm depth, 50 nL in total). We used either the Thy1-jRGECO1a or CBA mouse strain for AC-injected mice. Two V2L-injected mice were Thy1-jRGECO1a strain, and one was C57BL/6 strain. In this mouse and CBA mice, we additionally injected an AAV encoding for jRGECO1a (AAV2/

1-Syn-NES-jRGECO1a-WPRE-SV40, Addgene 100854-AAV1) in 3 locations in V1 (roughly the vertices of a 550 μm-wide equilateral triangle centered 3.5 mm posterior and 2.6 mm lateral to Bregma, 60 nL per location, 0.3 mm depth). AC and V1 coordinates in CBA mice were displaced posteriorly by 150 μm to compensate for the larger size of the brain relative to the Thy1-jRGECO1a mice (estimated from the bregma-to-lambda distance). To prevent backflow the pipette was kept in the brain for over 5 min after each viral injection. A circular craniotomy was performed over the left visual cortex (diameter: 3 mm; center: 3.5 mm posterior and 2.6 mm lateral to Bregma). The dura was left intact. An imaging window was constructed from three layers of microscope cover glass (1 x 5 mm, 2 x 3 mm diameter, Fisher Scientific, no. 1) joined with a UV-curable optical glue (NOR-61, Norland). The window was placed into the craniotomy and secured in place using black dental cement. A stainless steel headpost, specially designed to allow head fixation to the recording rig without interfering with pinna, was implanted into the skull with dental acrylic.

### Intrinsic signal imaging

One to two weeks after surgery and before starting two-photon imaging, intrinsic signal imaging was used to obtain a retinotopic map of the primary visual cortex. Mice were head-fixed and lightly anesthetized with isoflurane (1%) and injected intramuscularly with chlorprothixene (1 mg/kg). The eyes were coated with a thin layer of silicone oil (Sigma-Aldrich) to ensure optical clarity during visual stimulation. Optical images of cortical intrinsic signals were recorded using a Retiga QIClick camera (QImaging) controlled using Ephus[79] with a high magnification zoom lens (Thorlabs) focused on the brain surface under the glass window at 5 Hz. To measure intrinsic hemodynamic responses, the surface of the cortex was illuminated with a 620-nm red LED while drifting bar stimuli were presented to the right eye in a monitor. An image of the cortical vasculature under the window was obtained using a 535-nm green LED. Azimuth and elevation maps of the visual cortical areas were obtained by calculating the phase for each pixel of the discrete Fourier transform at the visual stimulation frequency[42,80]. The hemodynamic delay was canceled by subtracting the phase maps of the experiments from opposing moving stimuli.

### Two-photon imaging

Imaging was performed from 2.5 until 5 weeks post injection. All mice were imaged at <3.5 months of age, before the development of substantial high-frequency hearing loss[56,81]. We used a custom microscope (based on the MIMMS design, Janelia Research Campus, https://www.janelia.org/open-science/mimms) equipped with a resonant scanner, GaAsP photomultiplier tubes (10770PB-40, Hamamatsu) and a 16× (0.8 NA) objective (Nikon), controlled by ScanImage (Vidrio Technologies). O-rings were glued to the headpost, concentric with the center of the cranial window, to form a well for imaging. To prevent stray light from the LEDs from entering the imaging well, a flexible, conical light shield was attached to the headpost and secured around the imaging objective. GCaMP was excited using a Ti:sapphire laser (Chameleon Ultra II, Coherent) tuned to 920 nm. jRGECO1a was excited using a laser at 1070 nm (Fidelity, Coherent). We performed multi-plane imaging scanning in the axial direction using a piezo actuator (Physik Instrumente) to move the objective. For dual-color recordings, we recorded from GCaMP-expressing axons near-simultaneously at two depths in L1, and jRGECO1a at two depths in L2/3 of V1 (field of view [FOV] 80 × 80 μm, 512 × 512 pixels scanned at ~6 Hz, ~20–140 μm deep, 40 μm between planes). Alternatively, 2 planes (~160-200 μm deep, 40 μm apart) in V1 L2/3 neurons were first collected at ~10 Hz to map V1 neurons' RF before performing the GCaMP recording in L1 (4 planes at ~6 Hz, ~20-80 μm deep, 20 μm steps). To minimize cross-talk between the different calcium indicators when doing quasi-simultaneous recordings of GCaMP axons and L2/3 somata[82], the

1070 nm beam was turned off (<1 mW of power) when scanning in L1, and the 920 nm one was turned off when scanning in L2/3. For somatic GCaMP recordings, the FOV was either 480 × 480 μm or 320 × 320 μm, recorded at ~6 Hz (4 planes, 40 μm steps). Laser power measured after the objective was between 18-37 and 44-52 mW at 920 nm and 1070 nm, respectively. For each mouse, the objective optical axis was aligned perpendicularly to the imaging window.

Ambient noise in the sound-insulated two-photon recording cage was 39.0 dB SPL. During axonal (80 μm² FOV), and somata (320 μm² FOV) scanning, ambient noise was 39.2 and 42.5 dB SPL, respectively. dB SPL values were calculated within the mice hearing frequency range (2-80 kHz) and using $P_0 = 20$ μPa. By comparison, broadband white noise stimuli (2–80 kHz) reached the microphone placed at the location of the mice head at 48.0 ± 0.02 dB SPL, as expected from calculations based on sound attenuation in the air. Bandlimited white noise (2–20 kHz) reached the center of the speaker array at ~66 dB SPL.

### Speaker and LED array

The speaker and LED array was built in-house. Its design is available through the Champalimaud Hardware Platform (https://www.cf-hw.org).

Thirty-nine speakers (MULTICOMP PRO, # ABS-239-RC) and LEDs were mounted on a 3D-printed resin framework. Each position containing both a speaker and an LED was 10° apart in azimuth and 20° apart in elevation, thereby taking the shape of a spherical section spanning 120° in azimuth and 40° in elevation, with a radius of 18 cm.

The RGB LEDs were controlled using an LED controller board built in-house (HARP RGB LED; https://www.cf-hw.org/open-source-tools/electronic-devices/rgb-array). Light diffusers (5 mm-thick acrylic fitted in a 3D-printed frame, not represented in Fig. 1a) were positioned in front of each LEDs.

All speakers were regularly individually calibrated to flatten their frequency response using a Brüel & Kjær Free-field 1/4-inch microphone placed 2.5 cm from the speaker. The standard deviation of the power across frequencies for each speaker typically ranged from 0.97–1.57 dB (mean across speakers, 1.19 dB). Notice that, while the frequency response to white noise up to 80 kHz was flat after calibration (Supplementary Fig. 11), the speakers in the array are specified for frequencies up to 20 kHz. Consequently, they cannot produce pure tones of frequencies higher than 20 kHz without introducing harmonic distortions.

Sounds were digitized and amplified using a custom-built sound card and amplifier (192 kHz sampling rate, 24-bit precision; https://www.cf-hw.org/harp/harp-devices#h.p_lrxK9t3VI9s0)[83]. All the speaker-specific waveforms obtained from the calibration were stored in different channels of the sound card to minimize delay for sound stimulus presentation and applied for each speaker on a trial-by-trial basis.

The speakers playing the sound were switched 100 ms before sound presentation using three click-less audio switches (MAXIM, MAX4910; HARP audio switch https://www.cf-hw.org/open-source-tools/electronic-devices/audio-switch), controlling 12 to 15 speakers each.

HARP Audio switches, HARP RGB LED board, and HARP sound card were synchronized using a HARP clock synchronizer (https://www.cf-hw.org/open-source-tools/electronic-devices/clock-synchronizer).

We quantified reverberation in the speaker array. We applied a white noise pulse (20 ms, 2–80 kHz) and recorded the sound from a microphone placed in the center of the device. The power of the sound after offset decayed to half of the power in 3.7 ± 0.7 ms (exponential fit, mean ± s.d. across speakers, measure repeated three times). We fitted the decay of the log power of the sound after sound offset over time with a linear function and quantified the rate of this decay by the time taken for the reverberating sound to decay 20 dB (20−dB

reverberating time, RT20). RT20 was 19.4 ± 0.74 ms (mean ± s.d. across speakers, measure repeated three times). Similar values were obtained for the RT20 of the 2–20 kHz frequency band.

Stimulus sequences were generated using MATLAB (The Mathworks, Natick, MA). All the components in the 3D array were controlled using Bonsai[84].

### Measuring auditory and visual receptive fields

**Auditory stimulation.** Auditory stimuli were equalized white noise (bandlimited: 2–20 kHz or broadband: 2-80 kHz). Sound stimulation consisted of a continuous sequence of five bouts, each lasting 200 ms, with 10 ms raised-cosine onset/offset ramps ensuring smooth transitions. The entire sequence spanned a total duration of 1 s without any gaps between the bouts. Unless otherwise mentioned, bandlimited white noise was played at 85 dB SPL and reached the mouse head at ~66 dB SPL. Broadband white noise was played at 65 dB SPL and reached the mouse head at ~48 dB SPL.

**Visual stimulation.** LEDs were flashing white light at 3 Hz (50% duty cycle) at a luminosity of 11.4 cd/m².

**Stimulus sequence.** In the absence of visual stimulation, all the LEDs were set at a dim blue light level (0.25 cd/m²) to provide some background light. Auditory and visual stimuli were interleaved with a delay of 2 s from the offset of one stimulus to the onset of the next one. Auditory and visual stimulation positions were each randomized in a block design: the order of the 39 stimuli was randomized, and this was repeated 20 times (20 repetitions per trial type). Randomization sequences were independent between speaker and LED stimuli. Auditory and visual stimuli were presented 20 times at each position, resulting in 780 trials and an imaging session duration of 78 minutes. The magnitudes of the responses to the LED and speaker stimuli in AC boutons did not depend on the distance between successive stimuli (Supplementary Fig. 12).

In experiments measuring azimuthal RFs as a function of loudness levels (bandlimited noise), only speakers in the row at 0° of elevation were used (Supplementary Fig. 5). Loudness was varied on a trial-by-trial basis using the sound card. Each given stimulus position and loudness combination was repeated 15 times, resulting in an imaging session duration of 52 minutes.

**Audiovisual modulations of V1 neurons.** When measuring audiovisual modulations in V1 neurons, stimuli consisted of either an auditory, visual, or audiovisual stimulus. As before, these recordings were done with the LEDs set to provide dim blue light background illumination (0.25 cd/m²).

Auditory and visual stimulation consisted of five 100 ms-long bursts (10 ms raised-cosine onset/offset ramps for auditory stimulus) with a 50% duty cycle (1 s in total; Fig. 6a). In audiovisual trials, we ensured that auditory and visual stimuli were presented synchronously using a microphone and a photodiode. Auditory stimuli consisted of a bandlimited white noise (2–20 kHz, played at 85, 70 or 55 dB). Visual stimuli consisted of white light at luminosities of 2, 11.4, or 20.8 cd/m². We first mapped the population receptive FOV using pseudo-random sequences of LED flashes and identified retinotopic positions with RFs located at 30-60° in azimuth space and 0° in elevation. Subsequently, two adjacent LEDs located within the population receptive field center were used as targets to maximize visual responses of V1 neurons (the size of the FOV represents up to ~20° of azimuth angle) and 7 speakers per target LED were used. Positions of the speakers to the target LEDs were as follows ([azimuth, elevation]): [-40,0], [-20,0], [0,0], [20,0], [40,0], [0,-20], [0,20]. Each stimulus was repeated 7 times, resulting in an imaging session of 87 min. For analyses, the LED eliciting maximum response amplitude was determined for each cell, and all

quantification was performed using that LED and its associated speakers. Data from the two speakers non-congruent in elevation was not analyzed.

### Motorized speaker

For the experiments in Fig. 5 and Supplementary Fig. 9, a single loudspeaker (MULTICOMP PRO, # ABS-239-RC) was mounted on motorized rotation stage controlled by a stepper motor (8MR151 and 8SMC5-USB, respectively, Standa; speed: 40°/s, acceleration: 20°/s²) to present sound from −90 (left, ipsi-) to +100° (right, contra-lateral) around the frontal position (−90° to −30° with 20° step, −20° to +100° with 10° step), at a radius of 18 cm. Sound stimulus consisted of a broadband (2–80 kHz) white noise at 65 dB SPL. The waveform was a contiguous sequence of five bouts, each lasting 200 ms, with 10 ms raised-cosine onset/offset ramps ensuring smooth transitions. The entire sequence spanned a total duration of 1 s without any gaps between the bouts. Sounds were digitized and amplified using an Asus Xonar AE sound card (192 kHz sampling rate, 24-bit precision). The stimulus sequence was randomized in a block design: all the positions were randomized, and this was repeated 20 times. Stimuli were presented 20 times at each position, resulting in 320 trials and an imaging session duration of 69 minutes. The stimulus sequence as well as stimulus presentation and motor position were run using a Bpod (Sanworks) controlled using a custom Matlab code. These experiments were performed in the dark.

In 2 mice, the travel duration of the speaker varied across trials, with a maximum possible movement duration of 8 s. We thus used an inter-stimulus interval of 10 s to ensure that sound presentation occurred at least 2 s after the end of the speaker rotation. To control for the possibility that mice could estimate the speaker location from the traveling time of the rotating speakers between trials, in another 3 mice, we forced the speaker movements to be of equal duration regardless of the traveled angle. In these 3 mice, we added extra intermediate, random traveling locations so that the speaker rotated 180° (~8 s) in all trials. For instance, to go from −70° to −50°, the speaker will first go to 30° before coming back to -50°. Sound was presented 2 s after the end of the rotation. We observed similar location-specific sound-evoked responses in AC boutons in both protocols, showing that these signals were independent of the duration of the inter-trial speaker rotation time. Data was pooled across experimental conditions in Fig. 5.

### Pure tone stimuli

Frequency tuning from AC inputs to V1 was obtained in a separate design. A pair of loudspeakers (MULTICOMP PRO, # ABS-239-RC) were mounted at 2 cm from the left and right ear canals. Sine-wave pure tones were calibrated for loudness across frequencies. 500 ms-long, 55 dB pure tones were played at various frequencies from the right (contralateral) speaker. Sounds were digitized and amplified using an Asus Xonar AE sound card (192 kHz sampling rate, 24-bit precision).

### Histology

After completion of imaging experiments, mice were deeply anesthetized, transcardially perfused, and fixed overnight with 4% paraformaldehyde in 0.1 M phosphate buffer, pH 7.4. The brain was cut into 50-μm-thick coronal sections with a vibrating slicer (Leica). GCaMP was amplified using a polyclonal anti-GFP antibody (ThermoFisher, catalog #A-6455, diluted 1:4000) and an Alexa Fluor 488-conjugated secondary antibody (ThermoFisher, catalog #A-11008, diluted 1:1000) and counterstained with DAPI. Images were taken using a slide scanner (Zeiss AxioImager M2) connected to a sCMOS camera (Hamamatsu ORCA-Flash 4.0) with either a 10× (0.45 NA) or 20× (0.80 NA) objective (Zeiss). We did not observe any GCaMP-expressing V1 cell bodies in mice injected with AAVs expressing GCaMP6s or GCaMP8m in AC. We

used QuickNII and VisuAlign[85] to align the brain sections to the Allen Mouse Brain Atlas and identify the boundaries of cortical areas.

### Videography

We recorded videos of the contralateral mouse eye or face during two-photon imaging. We used a CMOS camera (Flea3 USB3 Vision, PointGrey) mounted with a telephoto lens (Navitar Zoom 7000) and a high-pass filter. Videos were recorded at ~20 Hz and the acquisition was synchronized with two-photon recording using a common TTL trigger sent by the HARP RGB LED board. The eye was imaged using scattered 920 nm light during two-photon imaging. Full-face movies were imaged using a high-power infrared LED (850 nm, Roithner LaserTechnick) illuminating the face. The eye was recorded in experiments measuring visual and auditory 2D spatial RFs using the speaker/LED array with bandlimited white noise (5/8 AC- and all three V2L-injected mice). No face recordings were obtained from these mice. The face was recorded in experiments measuring auditory 2D spatial RFs using broadband auditory white noise (2/2 Thy1-jRGECO1a and 6/6 CBA mice). For the azimuth and loudness tuning experiments, eye recordings were obtained from 2/3 mice and face recordings from 1/3 mice. For the motorized speaker experiments, eye recordings were obtained from 2/5 mice and face recordings from 3/5 mice. When measuring audiovisual modulations in V1 neurons, we obtained eye recordings from 4/8 mice and face recordings from 4/8 mice. Pupil was delineated post-hoc using DeepLabCut[86]. In ~200 video frames, eight points along the edge of the pupil were manually annotated (every 45°). Then a resnet-based convolutional neural network was trained to predict the location of these markers. The trained network was then manually evaluated and retrained by manually correcting labels from a subset of poorly predicted frames. The final network was then used to track the eight labels on every video frame. In video frames with average label placement probability > 0.9, with at least 6 markers with probability > 0.8, and where basic assumptions were not violated – for instance the top eye label should have higher y coordinates than the bottom one – an ellipse was fitted to the trustworthy labels and pupil area was calculated.

Motion energy from full-face videos and dimensionality-reduction in principal components using singular value decomposition were performed using Facemap[87].

### Two-photon data analysis

**Preprocessing of axonal data.** Motion correction and regions of interest (ROI) segmentation were performed using Suite2p[88]. We used the 'anatomical' parameter, such that suite2p placed ROIs over varicosities (likely presynaptic axonal boutons, referred to as boutons for simplicity) rather than axon segments. ROIs were manually added over boutons that were not picked up by Suite2p. Under the 'anatomical' detection, detected ROIs always encompassed single boutons. In a subset of experiments (5/8 AC-injected mice, 2–20 kHz white noise), ROIs were detected using the 'functional' parameter in suite2p, yielding detection of ROIs containing several boutons. We split responsive ROIs into (multiple) boutons using a watershed procedure and all the resulting single bouton ROIs were assigned the same $\Delta F/F$ trace.

We calculated $\Delta F/F = (F - F_0)/F_0$, where F is the average fluorescence of all pixels in each ROI and $F_0$ is the fluorescence of the ROI during the trial baseline.

Bouton ROIs were considered responsive if the strongest median response across all positions was different from the baseline (response time-window: 0.2 to 1.8 s from stimulus onset; two-sided Wilcoxon signed-rank test for paired sample test, $\alpha = 0.01$) and the response amplitude was larger than 0.15 $\Delta F/F_0$. Results were qualitatively identical when selecting responsive bouton ROIs with a bootstrapping method (strongest median response superior to 95% confidence interval of the baseline, and amplitude of response larger than 0.15 $\Delta F/F_0$).

**Comparison of azimuth and elevation modulation.** To compare how bouton responses depended on azimuth and elevation of the speaker, we downsampled azimuth positions to simulate a 40° x 40° isotropic speaker array. Azimuth positions were downsampled to all possible combinations of three positions covering 40° and 20° steps. The number of space-sensitive boutons was determined using a two-way ANOVA for each bouton and averaged across the 9 possible virtual arrays. Data was further averaged across positions of the same mouse and significant difference in the fractions of azimuth, elevation, and interaction-sensitive boutons was tested using a one-way ANOVA.

**Spatial Modulation Index (SMI).** SMI quantifies the relative proportion of the energy of the RF that could not be explained by the spatially averaged response and was calculated using Eq. (1) as previously described[89]

$$SMI = \frac{\sum_i^n (r_i - R)^2}{\sum_i^n r_i^2} \quad (1)$$

where $r_i$ is the average response to the stimulus in position i, n is the total number of positions and R is the average response to all stimuli.

**Decoding sound and light stimulus locations from axonal recordings.** To decode the location of the stimulus from the recorded bouton activity we used a naive Bayesian decoder. Because Bayesian decoding assumes independence of the data, we first grouped together the ROIs that belong to the same axon, using a correlation-based method[49]. In a subset of imaging sessions, we first identified boutons that had a visible axon shaft between them to calculate correlation values for boutons from the same axon. We determined that 58% of the boutons from the same axon had a Pearson's correlation value equal to or larger than 0.3 while less than 0.5% of the general population did. Based on this correlation value we selected all the pairs that were from the same axon and seeded a cluster with one randomly selected pair. The next randomly selected pair could be correlated with a coefficient over this value with only one of the ROIs with the existing cluster, in which case it joined the cluster, or not, in which case it seeded a second cluster. We iterated this procedure until all pairs were assigned. Activity in each cluster was assigned to the ROI with the largest mean $\Delta F/F$. We refer to these clusters as "axons". Receptive fields of the ROIs clustered together were largely overlapping. In sessions where this method yielded more than 10 axons, we applied a naïve Bayesian decoder approach.

For each selected session, we followed a 5-fold cross-validation procedure: in 5 runs, 80% of the trails (selected at random) were used to train the decoder, and decoding was performed in the remaining 20%, such that every single trial was decoded once. Decoding was performed as follows. The probability that the stimulus at a given trial was coming from the location s is given by Bayes' theorem:

$$P(s|n) = P(s) \times P(n|s)/P(n) \quad (2)$$

where n is the response of the entire axonal population. Assuming statistical independence of activity in the N axon recorded, we derived the likelihood function $P(n|s)$ as:

$$P(n|s) = \prod_i^N P(n_i|s) \quad (3)$$

where $n_i$ is the response of t1he axon i.

Responses were modeled using a normal distribution and thus, for all axon i and stimulus s:

$$P(n_i|s) = \frac{1}{\sigma\sqrt{2\pi}} e^{\frac{-(r-R)^2}{2\sigma^2}} \quad (4)$$

where, for axon $i$, $\sigma$ is the standard deviation of response to the stimulus s, r is the axon's response to the stimulus and R is the mean response to the stimulus s.

Therefore, the log-likelihood function is defined as:

$$\log P(n/s) = \sum_{i}^{N} \log(P(n_i|s)) \tag{5}$$

The decoded position is the maximum likelihood estimate $\hat{s}$, i.e., the position that maximizes the log likelihood.

$$\hat{s} = arg\,max \sum_{i}^{N} \log(P(n_i|s)) \tag{6}$$

The Euclidean distance between the actual and decoded position was then computed. Decoding errors were averaged over the decoded trials, runs, and positions of the same mouse to extract the decoding error per stimulus position (Supplementary Fig. 4a). In Fig. 3b, decoding errors were then averaged across stimulus locations. Decoding errors for the shuffle data were obtained using the same procedure, but for data where trial identity was permuted.

For calculating the decoding error relative to the V1 L2/3 somata population RF azimuth center (Fig. 4c), the decoding error was measured relative to the absolute difference between the azimuth location of the stimulus and V1's population RF azimuth center. Data was grouped in 10° bins.

For calculating the distance between the decoding error vs. number of axons used (Fig. 3c), we pooled axons from all the recording sessions and randomly drew a given number of axons without replacement (not to violate the assumption of data independence). The distance between the actual and decoded stimulus position was calculated on a single trial basis and averaged across trial types and runs. This procedure with random sampling was repeated 100 times to evaluate the mean and 95% confidence interval of the distance between decoded and actual stimulus position for a given number of axons.

To calculate the decoding error as a function of V1 retinotopic position from which the axons were recorded (Supplementary Fig. 4b), we randomly drew the $n_{min}$ axons (the minimal number of axons across all sessions) 10 times per session and averaged the decoding accuracy across resampling. We then calculated the moving average over 20° in azimuth in 5° steps.

**Best azimuth.** We selected reliable boutons to measure their spatial profile. Trials of each trial type were randomly split into two halves. Boutons were considered reliable if Pearson's pairwise correlation coefficient between the tuning curves obtained from the 2 halves was >0.3. The preferred (best) stimulus azimuth position was estimated from the responses averaged across all trials. The session's best azimuth was extracted by taking the median of the distribution of the bouton's best azimuthal position. This procedure was repeated 100 times to estimate the median and 95% confidence interval of the best azimuth per session. Only sessions with at least 10 reliable boutons were included. A 95% confidence interval of the linear fit was obtained from bootstrapping the residuals.

A similar analysis was conducted at the population level: responses were averaged across responsive boutons of one session and the population best azimuth was the azimuthal position eliciting maximum response amplitude (Supplementary Fig. 7).

**Spatial tuning curves.** The same procedure was applied to select reliable boutons. For each reliable bouton, the preferred stimulus position (maximum mean response across trials) was obtained from the first random half of the trials, and responses of the second random half were normalized to that maximum. Preferred azimuth was obtained by averaging responses across elevation, and vice versa for elevation. Boutons with the same preferred azimuth/elevation were pooled and normalized responses of the second random were plotted. This procedure was repeated 100 times to estimate the mean and 95% confidence interval of the tuning curves. Responsive boutons across all recording sessions from all mice were used.

**Frequency tuning.** A bouton was considered tone-responsive if stimuli at the frequency eliciting the largest responses resulted in significant responses (paired t-test between trial baseline and analysis window: from sound onset to sound offset + 500 ms, 1 s total, $\alpha = 0.01$ and if the magnitude of the mean response was >0.1 $\Delta F/F_0$). A bouton was frequency-tuned if it was tone-responsive and significantly modulated by frequencies (one-way repeated measure ANOVA, $\alpha = 0.05$). Frequency tuning curves were obtained by grouping boutons with the same best frequency and normalizing their responses to their best response.

**Onset and offset responsive boutons.** Response types were obtained using deconvolved calcium data. Median deconvolved activity from sound onset to 1 s after sound offset was used to determine the speaker eliciting the largest responses. Responses were categorized as onset or offset by comparing the event rate during the baseline period (1 s before sound onset), the sound presentation (1 s), or during a time window after the sound offset (1 s). Onset responses were identified as significantly increased activity between baseline and sound presentation period using a two-sided Wilcoxon signed-rank test for paired samples, ($\alpha = 0.01$). Offset boutons were identified as significantly increased activity between the baseline and offset periods and by activity not being different in the baseline and sound presentation periods. Decoder analysis was performed on $\Delta F/F_0$ data, using the categories obtained from deconvolved data. Analysis windows were confined to the period of sound presentation for ON boutons and to 1 s following the sound offset for OFF boutons.

**Population visual RF of V1 neurons.** To measure V1 L2/3 somata population receptive fields, full field-of-view responses were baseline subtracted (1 s to frame before LED stimulus onset) and averaged across repetitions. Responses during the stimulus window were time-averaged (1st frame after stimulus onset to 1st frame after stimulus offset, 1 s) to obtain a map of stimulus responses. The stimulus-response map R(az, el), where az and el are the coordinates azimuth and elevation, respectively, was fitted with a two-dimensional Gaussian to determine the RF location.

**Preprocessing of GCaMP somatic recordings.** For GCaMP6f somatic recordings (Fig. 6), motion correction and regions of interest (ROI) segmentation were performed using Suite2p, like for axonal data[88]. Data was manually curated. Only neurons with a mean trial baseline fluorescence signal stronger than the surrounding neuropil signal by more than 3% were kept[90] (79.2 ± 6.8%, mean ± sem, 8 mice). Fluorescence within each ROI was neuropil corrected using ($F = F - 0.7 \times Fneu$, where Fneu is the neuropil's fluorescence) and $\Delta F/F_0$ was then calculated using $F_0$ as the fluorescence during the trial baseline.).

For the analysis of the response magnitudes, neurons were selected if their best trial type elicited a significant response (paired t-test between baseline and response window, $p < 0.05$ and response magnitude averaged during the response window >0.05 $\Delta F/F_0$).

**Decoding sound location from somatic V1 recordings.** All neurons were included in the decoder analysis. A naïve Bayesian decoder was built from somatic recording data in each recording session. Speaker positions in individual AV or A trials were decoded using the maximum likelihood estimate as for axonal data (see above). We used a

leave-one-out cross-validation procedure and averaged the decoded distance to the actual speaker position in the AV trial over the 7 runs and across sessions and mice. Shuffle data was obtained by randomizing the trial labels without replacement 100 times.

## Reporting summary

Further information on research design is available in the Nature Portfolio Reporting Summary linked to this article.

## Data availability

The data that supports the findings described in this paper is available in Zenodo (https://doi.org/10.5281/zenodo.10685211)[91]. Raw two-photon recordings are available upon request. Source data is provided with this paper. The open source designs of the speaker and LED device and its associated circuits is available through our institute scientific hardware platform (Champalimaud Hardware Platform; http://www.cf-hw.org). Source data are provided with this paper.

## Code availability

Matlab code for analyzing the data and generating the figures is available at: https://github.com/camille-lab/ACaxonsSpace/.

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

## Acknowledgements
We thank Matthijs Oude Lohuis, Alfonso Renart, and all members of the Petreanu Laboratory for comments on the work, the Champalimaud Research Hardware Platform for technical support, and Adrien Jouary, Davide Reato, and Pedro G. M. Dias for help with experimental procedures and data analysis.

## Author contributions
C.M. and L.P. conceived the study. C.M. performed the experiments and data analysis. C.M. and M.B. performed surgeries and histology. C.M. and L.P. wrote the manuscript.

## Funding
This work was supported by a Marie Skłodowska-Curie individual fellowship (MSCA 798941; C.M.) and Human Frontiers Scientific Program fellowship (LT000064/2018-L; C.M.). L.P was supported by grants from Fundação para a Ciência e a Tecnologia (PTDC/MED449 NEU/30328/2017 and PTDC/MED-NEU/6645/2020), La "Caixa" Foundation (project codes LCF/PR/HR17/52150005 and HR22-00778) and by the Champalimaud Foundation. This work was supported by the research infrastructure CONGENTO, co-financed by Lisboa Regional Operational Programme (Lisboa2020), under the PORTUGAL 2020 Partnership Agreement, through the European Regional Development Fund (ERDF) and Fundação para a Ciência e Tecnologia (Portugal) under the project LISBOA-01-0145-FEDER-022170.

## Competing interests
The authors declare no competing interests.
