## [Peer Review File · Nature Communications]

Auditory cortex conveys non-topographic sound localization signals to visual cortexREVIEWER COMMENTS

Reviewer #1 (Remarks to the Author):

The manuscript by Mazo and colleagues address an important question, namely whether projections from auditory to visual cortex convey topographically organized information to V1. Several major recent publications investigated the existence and role of AC projections in V1, but the question of their topographical organization (essential to understand their role) had not yet been addressed. The conclusion that this topographic organization is not present will inform researchers aiming to study the role of crossmodal interactions in sensory systems. The paper is generally well done and well written, and the results support the conclusions of the authors. However, I have some concerns that the authors should address.

MAJOR

- 1) The authors claim that auditory-induced modulations are unlikely to be a factor underlying spatial auditory responses in AC axons. I think that their analyses support this claim. However, a question remains about the (albeit weak) spatial information present in V2L axons. As expected, most auditory-related information in V2L axons seems to have a behavioral origin. I believe that the authors should try to determine the source of this information. For example, is there any behavioral factor that could have been missed (it is known that behavioral modulations are much weaker in AC than in visual areas)? Or is this remaining spatial information coming from AC, and is then indirectly relayed to V1?
- 2) I believe that the use of a rotating arm presents a confound, unless it can be proven that the mice could not predict where the rotating arm was. It is true that the results are comparable to those obtained with the speaker array (Fig. S6), but I wonder why the authors did not just implement a relative rotation of the mouse and the speaker array, or extend the speaker array. As a minimum, the authors need to precisely describe the procedure meant to ensure that mice could not spot or predict the location of the rotating arm (e.g. when was it moved relative to the onset of sound presentation? How did the authors ensure that the location of the speakers could not be estimated by earlier auditory cues and/or by visual cues?).
- 3) The last section of the results needs, in my opinion, to be carefully re-assessed:
 - a. It is unclear whether auditory stimuli were presented alone. In some parts of the texts it seems so (this is supported by stats), but not in the figures.

b. The authors mention that several stimulus intensities were shown, but the relative data is lacking. The authors do say that the enhancement did not change as a function of intensity, but one wonders if auditory or visual responses on their own varied as a function of loudness. In the field of multisensory integration enhancement is normally dependent on the strength of unisensory responses, so I wonder how this assessment of done. Did the authors consider absolute changes in dF/F ? Or relative changes (e.g. relative to the higher unisensory response)? This is an essential computation to support the authors' claim.

c. Could the speaker location be decoded from Auditory (and not AV) responses of V1 neurons? I think that this is a crucial element to investigate the role of auditory spatial information sent to V1.

d. Most importantly, the authors did not control for any behavioral-induced factor mediating the apparent auditory modulation of V1 responses. In fact, the AV modulation results, in the absence of any manipulation of either AC projections or movement-related modulatory signals, cannot in my opinion be conclusively ascribed to AC projections. Any increase in V1 responses in the presence of sound is in fact likely to be primarily a consequence of behavioral modulation, as shown by previous studies. In particular because the stimuli employed here did not induce discernible behavioral differences, I would say that this is compatible with the reported effects.

In general, I believe that the authors need to either support their claims with experiments capable of disentangling direct auditory vs behavioral modulation of V1, or perhaps exclude this section from the main results of the manuscript. I believe that the manuscript would be equally informative and relevant for the community.

MINOR

1) In general the authors should carefully check figure numbers. I noted some inconsistencies here, but several others may be present throughout the text.

2) Line 90: There is an extra comma that should be removed

3) Line 191: While movement-induced modulations of neuronal activity are widely found across cortical areas, these are not similarly present everywhere. For instance, see the surprising absence in auditory cortex reported in oude Lohuis et al. (2023, biorXiv).

4) Section on spatial auditory responses: The authors refer to Supplementary fig. 5 in lieu of Supplementary Fig. 4.

5) Line 204: it should be spatial and not spatially

6) Line 252: What are the two ipsilateral locations? Should these be contralateral? And where can this be seen in Fig. 2f? Did the authors simply mean that AC boutons have preferred spatial locations?

7) The rotating arm section of the methods refers to fig. 7, which is not existent.

8) The authors state “We presented interleaved auditory white noise bursts (2-20 kHz) and flashing white LED light (Figure 1c).” What is the interstimulus interval (ISI?) - is the duration randomized? Is the LED light always following the auditory stimulus (e.g. predictable)? How many trials do animals on average do per session? And what is the total number of samples in each individual location per session (i.e. are they the same for both orientations, azimuth ?) (The same applies to both speaker array and movable speaker)

9) How is the sequence for the locations for individual stimuli selected? A greater distance between successive stimuli might be more salient (and potentially induce saccades and larger corollary discharges). Is there any effect when the distance between successive stimuli is larger? (The same applies to both speaker array and movable speaker)

10) Fig. 1f shows that, while a large part of the connections is sending projections from AC and V2L to layer 1 (L1) of the visual cortex, the targets in deeper layers of the visual cortex are also clearly evident by the fluorescent data (figure 1f). The authors should perhaps better discuss if it is possible that different results might have been obtained by focusing on projections to different V1 layers (if e.g. these came from different sources in A1). This focus on l1 could also be made evident in the title or abstract.

11) Viral AC injections: It is unclear how this second injection site (3.0 mm, 4.7 mm 0.5 mm depth) would hit the auditory cortex. Why did you choose this dual injection strategy?

12) “Sound stimulation consisted of five 200 ms bursts with 10 ms raised-cosine onset offset ramps (1s total)”. This is not entirely clear - what’s the difference with a constant 1 s white noise burst?

13) "For the analysis of the magnitude of the audiovisual modulation, we excluded neurons with high variability in their response (s.d of the response > 10, <1 % of all neurons)." Variability to a particular stimulus, or variability over the entire session? What is the reason for this exclusion?

Reviewer #2 (Remarks to the Author):

It has been noted for a long time that auditory stimuli elicit responses in the mouse primary visual cortex (V1). However, the nature of the auditory information conveyed to V1, its organization, and its role in visual processing remain unclear. Recent evidence has shown that part of the response of V1 neurons to auditory stimuli is caused by uninstructed behavioral responses rather than the auditory stimuli themselves.

Projections from the primary auditory cortex (AC) are known to reach V1. This study aimed to investigate whether sound location information is present in AC projections to V1 and whether it is topographically organized, similar to the superior colliculus. Dual-color two-photon microscopy was employed to image calcium responses of AC boutons (and V2L boutons) in V1 and relate them to the responses of V1 neurons. The study found that AC boutons in V1 were often modulated by sound location, but the spatial extent of their responses varied greatly. As a control, the activity of V2L boutons in V1 was also imaged, revealing more confined visually induced responses, while auditory responses in these boutons were not spatially confined.

Contrary to V2L, AC boutons did not exhibit a topographic organization matching visual space in V1. However, sound location could be decoded based on AC bouton responses, albeit with better accuracy for azimuthal space than for elevation. In contrast, location decoding was not possible based on V2L bouton activity in response to sounds. The authors did not find evidence linking AC location responses to uninstructed behavior of the mice. Although AC activity often correlated with pupil size and facial movements, there was no relationship between facial movements and sound location, nor could sound location be decoded from V2L boutons.

Furthermore, the study assessed whether the responses of V1 neurons were more strongly modulated by sounds when the location of the sound and visual stimulus (or their brightness and loudness) were congruent, but no evidence supporting this hypothesis was found.

Taken together, this is a thoroughly executed study providing strong evidence for location-specific auditory input to V1 for the first time. The statistical analyses conducted were appropriate. However, it

remains unknown how location-specific auditory information is actually used in V1 and what its behavioral relevance is.

Several issues still need attention.

Major issues:

1. In the Methods section, it is mentioned: "We recorded videos of the mouse eye/face during two-photon imaging." Please specify which data was available/analyzed for each group since it appears that in some sets of animals, only the eyes were imaged, while in another set of animals (the CBA mice only?), full-face movies were recorded.

2. To strengthen the claim that location-modulated responses of AC boutons are not related to uninstructed behavior of the mice, it would be useful to show that location-specific responses are also present in trials without facial movements.

3. It is unclear whether the modulation of V1 responses by AC is directly caused by auditory inputs to V1 or indirectly by uninstructed behavior. To address this, it would be valuable to determine if the same result is obtained when only trials without facial movements are used or when facial movements are regressed out (if this data is available, please refer to point 1).

4. In line 232, the authors state that Thy1-jRGECO1a mice were on a C57bl/6 background. Please provide more information about how the mouse line was maintained.

5. It would be useful to provide information regarding whether location-specific suppression of spontaneous activity was observed in the dark (considering Deneux et al., eLife 2019).

Minor issues:

1. There are several typos in the text. Please check lines 90, 198, 204, 310, 322, 329, 330, 343, 344, 384, 609, 623, 733.

2. The authors refer to Supplementary Figure 5, but it should be Supplementary Figure 4.

3. The sentence on line 310 is a bit confusing due to the mention of the lack of selectivity: "As in the AC as a whole, AC afferents were spatially non-selective, or selective to varying degrees to sounds in specific ipsi- or contralateral locations."

4. The authors sometimes refer to responses of boutons and sometimes of axons, which does not seem consistent. Please review and fix where necessary.

5. More information and analysis of onset and offset responses separately would be helpful. Are onset responses dominant, as suggested by Deneux et al.?

6. The stimulus parameters are confusing (e.g., how can a 1-second stimulus consist of 5 200 ms bursts?). Please provide clearer descriptions.

7. In line 111, the authors state: "We confirmed that AC boutons were frequency-tuned, as expected from auditory responses in AC neurons (Supplementary Figure 2a)." However, only three examples are shown, and no quantification is provided to support this statement.

8. Please ensure that the statistics used are always clear and complete. Statements like "The position eliciting the largest response varied across AC axons, with a slight enhancement in boutons tuned to high-azimuth positions (Fig. 2g)" (line 152) need to be supported by statistics.

9. In line 189: "First, these modulations are low dimensional, i.e., they affect all neurons similarly, while the spatial profile of the sound-evoked activity in AC to V1 inputs was diverse across boutons, even within the same session (Figure 2a)." It seems to be referring to the wrong figure or panel.

10. Please adjust the scales in Supplementary Figure 4b and 4c so that the data can be assessed more easily.

Reviewer #3 (Remarks to the Author):

In this carefully executed study, Mazo et al. have managed to record simultaneously in mouse primary visual cortex the inputs coming from auditory cortex and the activity of V1 neurons while playing sounds and visual stimuli arriving from different spatial locations. Thanks to these experiments they could show that spatial information about the sound reaches visual cortex but is not spatially aligned with retinotopic information. This result is important, as it indicates that alignment of visual and auditory spatial receptive fields, at least in mice, is specific to the superior colliculus. It is of course not possible to exclude that it is a mouse-specific result, potentially related to the poor accuracy of spatial hearing in mice, however this result is extremely useful for understanding the mechanisms underlying in orienting behaviors based on multisensory cues.

I recommend nevertheless a few verifications in order to make sure that weak RF alignment has not escaped the careful analysis of the data and to evaluate the generality of this results.

1/ If I understood well, the authors have evaluated the spatial alignment of receptive fields with two methods. The direct one compares the averaged best azimuth of a AC axonal population against the average best azimuth for visual stimuli for the underlying V1 neurons. This method is likely to be highly sensitive to response variability, as the best azimuth of a given axon is not measured with high certainty when the axon itself is not sharply tuned. Particularly, the contribution of weakly tuned axons may increase variability and hide local population tuning. To verify that it is not the case, I recommend to perform the analysis in the opposite order. First average responses across all axons and then compute the preferred azimuth for comparison with V1 neurons. This population measure would certainly complement the classifier analysis used by the authors, which is supposed to address the point, but may do this less accurately and directly than the method I propose, due to the sensitivity of classifiers to noise in high dimensions.

2/ Earlier studies have suggested that auditory inputs to V1 depend strongly on intensity. Here the intensity used 66dB for C57 mice and ~50 dB for CBA mice is rather in the low range. If the authors have an indication whether the lack of spatial RF alignment is still seen at higher intensity (80dB in C57 or 70 dB in CBAs), it would significantly strengthen their results.

3/ Line 351 in the discussion, the authors state that V1 does not send inputs to AC. This statement should be mitigated. Connectivity studies show clearly an asymmetry but do not fully exclude weak inputs from V1 to AC. Functional studies show clear visual responses in deep layers of AC, e.g. the recent work of the Hasenstaub lab.

4/ Two suggestions to be included in the discussion. As the author mentioned a recent study show that spatial information about sounds is more efficiently extracted with high frequency information. The

Kanold lab has demonstrated few years ago that despite have much better high frequency hearing CBAxC57 mice still have weak high frequency representations in cortex.

<https://www.nature.com/articles/s41598-020-67819-4> It may explain why the auditory visual spatial mapping does not emerge in mice at the cortical level. It may be that in the lab mouse usual sound scape (small cage likely with reverberations), the directionality information carried by lower frequency is quite unreliable, preventing alignment, while the more local high frequency are more reliable.

Another idea in the same direction, given the poor accuracy of sound localization in mice it may not be advantageous to add this information to the already accurate visual representation of events in space.

Reviewer #4 (Remarks to the Author):

Although projections from auditory to primary visual (V1) cortex are well established, the role of these projections remains a significant gap in our understanding of audiovisual integration. Here, Mazo et al convincingly test the hypothesis that auditory projections to V1 are topographically matched, and establish that this is not the case: although axons from auditory cortex did demonstrate spatial selectivity, this was not correlated with the receptive field of the colocalised population of V1 neurons. To demonstrate that their labelling technique and analyses could identify sensory-map alignment if it was present, the authors reproduced the result that spatial responses of V2L-axons do correspond to the spatial selectivity of the colocalised V1 population. The authors further demonstrate that the signals from auditory cortex are only weakly lateralised, and that spatial coherence between auditory and visual stimuli does not significantly impact the auditory modulation of V1 activity. The authors convincingly deal with concerns regarding audio-evoked movement signals.

These data represent an important, and original, contribution to the field. Aside from the issues (one major, several minor) listed below, I found the methodology and analyses convincing, and believe the central thesis of the paper is supported by the data.

Major concerns:

- 1) In several experiments (e.g. Fig. 4b), the authors claim to play sounds of “2-80 kHz auditory white noise” with their speakers (MULTICOMP PRO, # ABS-239-RC). However, these speakers are only rated to deliver sounds up to 20 KHz. Typically speakers that can reliably produce frequencies at the top end of this range cost hundreds of euros. Therefore, speaker calibration plots, including the accurate reproduction of sinusoids in this high-frequency range, need to be provided as supplemental data to demonstrate that these speakers are capable of delivering the sounds that the authors claim to produce.

Minor concerns:

1) Although I do believe the result that projections from auditory cortex are spatially selective, it would strengthen this finding to show that it was not specific to the type of auditory stimulus presented (e.g. showing that within a session, the spatial selectivity of a neuron is not highly dependent on the frequency spectra). This would reassure the reader that differences in response to not reflect differences in the speakers themselves which may remain even after calibration.

2) First sentence of abstract: "Perception requires binding spatiotemporally congruent multimodal sensory stimuli." One can perceive the world without binding stimuli. The first sentence of the intro is also too general (many animals don't have multiple specialised sense organs).

3) In figure legends (e.g. Fig 2d) it reduces readability to report all p values exactly and state the comparisons. When highly significant, I would suggest stating e.g. *** = $p < 0.001$ at the end of the figure.

4) Fig 3b why is a threshold used instead of simply showing the distance of the estimate from the sound location?

5) Related to the above, in Supplemental Figure 3a, I am concerned about the accuracy of the shuffle. The "chance level" decreases at -20/100 degrees (presumably because there are less "correct" guesses at these locations because of the thresholding). However, why does the shuffle accuracy show no corresponding decrease? Also, why is the shuffle more accurate than chance for all locations in the V2L axons of this figure?

6) Related to the above, in Supp. Fig 5c, why are the chance levels different from 3a?

7) "We scored the accuracy of the decoder by measuring the fraction of times that the decoded position with the largest likelihood was within 10° of the actual one." This is only true to azimuth--not elevation--I believe? But this isn't stated in the main text.

8) "with a only slight , but significant, overrepresentation of contralaterally-tuned ones (Figure 5c,d)." Unless I missed it, there is no mention of this significance, or the significance test, in the figure?

9) Sup. Fig. 2a: does the changing (white to black) on the frequency bar at the bottom mean anything? If not, I would remove this bar completely, especially as a thickening bar is more associated with loudness/intensity than frequency.

10) Missing references:

a. [https://www.cell.com/neuron/fulltext/S0896-6273\(17\)30007-7](https://www.cell.com/neuron/fulltext/S0896-6273(17)30007-7) where activation of AC > V1 projections did not result in suppression of V1 (in contrast to Ref 15 in the ms).

b. <https://pubmed.ncbi.nlm.nih.gov/37295419/> where visual cortex inactivation doesn't impact auditory spatial localization in mice.

c. <https://pubmed.ncbi.nlm.nih.gov/32402272/> where higher visual regions are suggested to more closely reflect audiovisual detection behaviour.

11) There are a lot typos in the manuscript. I have noted several here, but have likely missed others. It warrants a careful read before publication:

a. Line 35 "stablished"

b. Fig2a: should not be "across" individual mice.

c. Line 679 "was" should be "were"

d. Line 204 "their spatially pattern in ACàV1 boutons"

e. Line 224 refers to "LM" for the only (I think) time in the paper

f. Line 262 "with a only slight"

g. Line 311 "specific psi- or contralateral locations"

h. Line 325 "a large fraction of AC inputs was"

i. Line 326 "for ipsilaterally sounds originating"

j. Line 330 "(2-80 kHz white noise"—no closing bracket

k. Line 333 "with an accurate spatial map as V1"

l. Line 337 "enhanced by spatially congruent bimodal stimulus"

m. Capitalization of axes labels are inconsistent throughout the ms/supp

REVIEWER COMMENTS

Reviewer #1 (Remarks to the Author):

The manuscript by Mazo and colleagues address an important question, namely whether projections from auditory to visual cortex convey topographically organized information to V1. Several major recent publications investigated the existence and role of AC projections in V1, but the question of their topographical organization (essential to understand their role) had not yet been addressed. The conclusion that this topographic organization is not present will inform researchers aiming to study the role of crossmodal interactions in sensory systems. The paper is generally well done and well written, and the results support the conclusions of the authors. However, I have some concerns that the authors should address.

We thank the reviewer for the insightful comments and constructive suggestions.

MAJOR

1) The authors claim that auditory-induced modulations are unlikely to be a factor underlying spatial auditory responses in AC axons. I think that their analyses support this claim. However, a question remains about the (albeit weak) spatial information present in V2L axons. As expected, most auditory-related information in V2L axons seems to have a behavioral origin. I believe that the authors should try to determine the source of this information. For example, is there any behavioral factor that could have been missed (it is known that behavioral modulations are much weaker in AC than in visual areas)? Or is this remaining spatial information coming from AC, and is then indirectly relayed to V1?

We agree with the reviewer, that weak sound location-specific activity in V2L axons might originate from either location-specific behavioral responses that we could not detect or, indirectly, from AC inputs to V2L.

Using motion energy from facial videos — a method successfully used in previous literature to decode sound identity (Bimbard et al., Nat Neurosci 2023) and proven to be far more sensitive and reliable than the acoustic startle reflex (Clayton et al., biorxiv 2023) — we were still unable to decode sound location. We thus favor the hypothesis that spatial information in V2L inputs is inherited from the AC.

However, experimentally establishing the role of the AC in the weak sound localization signals of V2L inputs in V1 poses a considerable challenge. These signals are exceedingly faint, remain inconspicuous when examining individual animals, and only become apparent when aggregating boutons across multiple animals during the decoding analysis. Moreover, we feel that doing so would be out of the scope of this paper. We measured sound responses in V2L inputs mainly as a control to the specificity of the location-specific responses in AC inputs. Thus, the paper being about AC inputs to V1, we feel that dissecting the mechanism of the spatial responses in V2L inputs to V1 would be out of the scope of this work.

We now directly address the weak sound spatial information present in V2L inputs in a dedicated paragraph of the discussion.

Lines 359-369: “On the contrary, inputs from V2L contained less information about the location of sounds than AC inputs, as expected from a visual area. However, sound location could be decoded slightly above chance when integrating V2L axons from multiple sessions and animals (Figure 3c). The origin of the weak sound localization signals in V2L axons is unclear. Since sound-evoked responses in these axons exhibit stronger correlations with behavior than those originating in the auditory cortex (Supplementary Figure 6d), as also observed in somatic recordings²⁹, these signals may indicate modulations linked to location-specific uninstructed movements. Nevertheless, it’s important to note that sound location could not be decoded from facial movements and pupil dilations, which does not support this hypothesis. Hence, given the proximity of the two cortical areas⁶², the weak sound localization signals in V2L inputs in V1 are likely inherited from sound localization signals originating from the AC.”

2) I believe that the use of a rotating arm presents a confound, unless it can be proven that the mice could not predict where the rotating arm was. It is true that the results are comparable to those obtained with the speaker array (Fig. S6), but I wonder why the authors did not just implement a relative rotation of the mouse and the speaker array, or extend the speaker array. As a minimum, the authors need to precisely describe the procedure meant to ensure that mice could not spot or predict the location of the rotating arm (e.g. when was it moved relative to the onset of sound presentation? How did the authors ensure that the location of the speakers could not be estimated by earlier auditory cues and/or by visual cues?).

Our primary rationale for conducting repeated measurements of auditory spatial receptive fields, utilizing a speaker affixed to a rotating arm, was to address the potential influence of sound localization signals caused by discrepancies in the sounds emitted by distinct speakers within the array. (This concern was raised by Reviewer 4).

We now revised the text and included a paragraph in the results making this more clear

Lines 257-268: **“Spatially specific sound responses in ACàV1 inputs are not due to differences across speakers.**

We wondered if, despite having calibrated the frequency responses across speakers, the spatially specific sound responses of AC axons could still reflect unnoticed variations in the sounds produced by the different speakers in the array. To ensure that this was not the case, we mounted a single loudspeaker on a rotating arm and presented sounds at different azimuthal positions (Supplementary Figure 9a). We confirmed that the spatially specific sound responses of AC axons obtained with the speaker array were similar to those measured using the rotating speaker, as assessed by their SMI and the performance in decoding azimuthal position (Supplementary Figure 9b,c). Furthermore, like when using the speaker array for stimulation, the spatial auditory responses in ACàV1 inputs lacked topographical organization when evaluated using a rotating speaker (Supplementary Figure 9d).”

Furthermore, our speaker array encompassed only 120 degrees, thereby providing a restricted scope for investigating the relative abundance of boutons tuned to ipsilateral and contralateral locations. Given that the rotating speaker allowed probing a wider range of positions around the animal's head we used the rotating speaker also for these measurements.

We believe it is unlikely that the sound localization signals in AC axons relate to the localization of the speaker position using earlier sound cues, given the stimulation protocol:

The sequence of positions was randomized, and the experiments were conducted in a dark environment. The interstimulus interval (ISI) was set to 10 seconds to allow the speaker to attain its final position at least 2 seconds before the onset of the stimulus.

While stimulus positions located at the boundaries of the sampled space resulted in longer durations of movement on average, complete differentiation based solely on travel time occurred only in instances when the speaker journeyed between the two utmost positions (-180 and +180 degrees) – a distinction that was apparent in just 1 out of 17 stimuli.

Furthermore, attempting to estimate positional cues by analyzing the duration of the rotating motor's noises across numerous trials would demand significant cognitive exertion. It's important to note that the mice were neither trained nor rewarded for such a task.

In any scenario, to definitively determine whether the sound location signals in the auditory cortex (AC) were connected to the mice's capacity to gauge the speaker's position through preceding travel time durations, we conducted supplementary experiments. In these experiments, the speaker rotation duration was forced to be identical in each trial, independently of the angular distance between consecutive trials.

As before, we found spatially tuned AC boutons representing both ipsi- and contralateral locations and included them in Figure 4.

We now describe these new experiments with fixed inter-trial speaker rotation duration in the methods. We also expanded the methods sections to precisely describe the procedures to rotate the speaker across trials.

Lines 670-680: "In 2 mice, the travel duration of the speaker varied across trials, with a maximum possible movement duration of 8 s. We thus used an inter-stimulus interval of 10 s to ensure that sound presentation occurred at least 2 s after the end of the speaker rotation. To control for the possibility that mice could estimate the speaker location from the traveling time of the rotating speakers between trials, in another 3 mice, we forced the speaker movements to be of equal duration regardless of the traveled angle. In these 3 mice, we added extra intermediate, random traveling locations so that the speaker rotated 180° (~8 s) in all trials. For instance, to go from -70° to -50°, the speaker will first go to 30° before coming back to -50°. Sound was presented 2 s after the end of the rotation. We observed similar location-specific sound-evoked responses in AC boutons in both protocols, showing that these signals were independent of the

duration of the inter-trial speaker rotation time. Data was pooled across experimental conditions in Figure 5.”

3) The last section of the results needs, in my opinion, to be carefully re-assessed:

a. It is unclear whether auditory stimuli were presented alone. In some parts of the texts it seems so (this is supported by stats), but not in the figures.

We modified the figure to include, and make more clear, that auditory stimuli were also presented alone. We also modified the text to make this more clear.

b. The authors mention that several stimulus intensities were shown, but the relative data is lacking. The authors do say that the enhancement did not change as a function of intensity, but one wonders if auditory or visual responses on their own varied as a function of loudness. In the field of multisensory integration enhancement is normally dependent on the strength of unisensory responses, so I wonder how this assessment of done. Did the authors consider absolute changes in dF/F ? Or relative changes (e.g. relative to the higher unisensory response)? This is an essential computation to support the authors' claim.

We did show stimuli of different brightness and loudness. We now show how V1 responses to light flashes and sounds on their own change with brightness and loudness in Supplementary Figure 10. In the previous version, we assessed only absolute changes in dF/F (AV-V). We now also measured relative changes (AV/V) and doubled the number of mice with new experiments. We tested if the AV modulations depended on specific brightness and loudness levels using 3-way ANOVA using, brightness, loudness, and speaker positions as factors. The ANOVA analysis detected an interaction between loudness and brightness when using relative modulations, but not when using absolute. However, both when considering absolute or relative changes in dF/F the ANOVA analyses did not find speaker position as a significant factor or any interactions with the stimulus strength.

We now mention this in the text. We also added a new Supplementary Figure 10, where we show unisensory and AV modulations as a function of stimulus intensity.

c. Could the speaker location be decoded from Auditory (and not AV) responses of V1 neurons? I think that this is a crucial element to investigate the role of auditory spatial information sent to V1.

We added these analyses to Figure 6 (panels b-d) and Supplementary Figure 10 (panels b,d). As with AV responses, we did not observe any modulations of the A-only responses as a function of the distance to the V1 neurons' visual RF. We could not also decode speaker location from the V1 neurons' responses to A-only stimulation.

d. Most importantly, the authors did not control for any behavioral-induced factor mediating the apparent auditory modulation of V1 responses. In fact, the AV modulation results, in the absence of any manipulation of either AC projections or movement-related modulatory signals, cannot in my opinion be conclusively ascribed to AC projections. Any increase in V1 responses in the presence of sound is in fact likely to be primarily a consequence of behavioral modulation, as shown by previous studies. In particular because the stimuli employed here did not induce discernible behavioral differences, I would say that this is compatible with the reported effects.

In general, I believe that the authors need to either support their claims with experiments capable of disentangling direct auditory vs behavioral modulation of V1, or perhaps exclude this section from the main results of the manuscript. I believe that the manuscript would be equally informative and relevant for the community.

We now show that AV modulations are of equal magnitude regardless of the amplitude of the facial motion energy. We also isolated sessions with no significant increase in facial motion energy upon sound stimulation and found that AV modulations are still present and that the speaker position still did not affect their magnitude. These analyses show that the AV responses cannot be fully explained by behavioral modulations, as suggested by Oude Lohuis et al., *bioRxiv* 2022, Williams et al., *bioRxiv* 2022 and Deneux et al. *eLife* 2019 and others. However, we agree with the reviewer that the contribution of AC projections in these signals remains unclear. While this certainly is a limitation of the experiments, we believe that the results are still valuable, as they show that sound-induced modulations, regardless of the degree of involvement of AC projections, do not contain information about the spatial congruence of the AV stimuli. They also show that A-only responses bear no relation with the retinotopic preference of the V1 neurons. This is in contrast to earlier reports in cats (i.e. Morell et al. *Nature* 1972) that remained untested in mice for decades. As a consequence, we decided to leave the experiments as a main figure. However, we now make clear these limitations when discussing the observations.

Lines 319-332: "Sound elicited unstructured movements in a loudness-dependent manner. Loud sounds evoked reliable movements while quieter ones did not (Supplementary Figure 10e,f,h). In contrast to this, the magnitude of the AV modulation was similar across loudness levels (Supplementary Figure 10d). V1 neurons showed AV modulations and A-only responses at the lowest sound intensity even though sound-evoked movements were undetectable. Auditory responses and AV modulation remained independent of the distance between the speaker position and the neuron's RF or visual stimulus, respectively, when using quiet sounds (Supplementary Figure 10g,i). This suggests that some components of the auditory-evoked activity and the sound modulations of the visual responses in V1 are motor-independent, as suggested previously 14,29. Whether these motor-independent signals depend on AC→V1 inputs remains to be determined. While we cannot completely dismiss the possibility that certain components of the signals may contain information regarding the spatial congruency of audio-visual stimuli or the positioning of sounds relative to the receptive fields of V1 neurons, these signals do not exert a predominant influence on V1 responses in mice when they are passively exposed to sounds."

We also now include the extra analyses on the contribution of face motion to the V1 signals in Supplementary Figure 10f-i

MINOR

1) In general the authors should carefully check figure numbers. I noted some inconsistencies here, but several others may be present throughout the text.

We fixed inconsistencies in figure numbers.

2) Line 90: There is an extra comma that should be removed

We fixed this sentence.

3) Line 191: While movement-induced modulations of neuronal activity are widely found across cortical areas, these are not similarly present everywhere. For instance, see the surprising absence in auditory cortex reported in oude Lohuis et al. (2023, biorXiv).

We now explicitly mention that movement-related modulations were shown in previous work to be largely absent in AC, which strengthens our argument that sound localization signals in AC→V1 inputs are independent of uninstructed movements.

4) Section on spatial auditory responses: The authors refer to Supplementary fig. 5 in lieu of Supplementary Fig. 4.

We corrected this error.

5) Line 204: it should be spatial and not spatially

We corrected this typo.

6) Line 252: What are the two ipsilateral locations? Should these be contralateral? And where can this be seen in Fig. 2f? Did the authors simply mean that AC boutons have preferred spatial locations?

We mean that a fraction of the AC boutons had a peak azimuth in one of the two ipsilateral positions sampled with the array (-20 and -10 degrees in Figure 2g). We now made this more explicit:

Lines 272-273: "Many AC boutons in V1 exhibited peak responses when sound was presented in one of the two ipsilateral locations in our speaker array (-20° and -10° in azimuth in Figure 2g)."

7) The rotating arm section of the methods refers to fig. 7, which is not existent.

We corrected this error.

8) The authors state “We presented interleaved auditory white noise bursts (2-20 kHz) and flashing white LED light (Figure 1c).” What is the interstimulus interval (ISI?) - is the duration randomized?

2 s separated the stimulus offset of one modality from the stimulus onset of the other modality. The ISI was not randomized. This is now made clearer in the methods section:

Lines 626-627: “Auditory and visual stimuli were interleaved with a delay of 2 s from the offset of one stimulus to the onset of the next one.”

Is the LED light always following the auditory stimulus (e.g. predictable)?

LED and speaker stimulus presentations were interleaved with a fixed ITI, but the position was not predictable: stimulus location sequences were random.

We now made this more explicitly in the methods section:

Lines 625-628: “Auditory and visual stimulation positions were each randomized in a block design: the order of the 39 stimuli was randomized, and this was repeated 20 times (20 repetitions per trial type). Randomization sequences were independent between speaker and LED stimuli.”

How many trials do animals on average do per session? And what is the total number of samples in each individual location per session (i.e. are they the same for both orientations, azimuth ?) (The same applies to both speaker array and movable speaker).

We added this information to the methods section:

Speaker and LED array:

Lines 628-630: “Auditory and visual stimuli were presented 20 times at each position, resulting in 780 trials and an imaging session duration of 78 minutes. “

Motorized speaker:

Lines 664-667: “The stimulus sequence was randomized in a block design: all the positions were randomized, and this was repeated 20 times. Stimuli were presented 20 times at each position, resulting in 320 trials and an imaging session duration of 69 minutes“

9) How is the sequence for the locations for individual stimuli selected?

The sequence for the location was randomized in a block design: the order of the 39 stimuli was randomized, and this was repeated 20 times (20 repetitions per trial type) — the same applies

to the movable speaker. The same randomization procedure was applied for the LED sequence, and the randomization was independent of that of the speakers. We now made this clearer in the methods:

Speaker and LED array:

Lines 625-627: “Auditory and visual stimulation positions were randomized in a block design: the order of the 39 stimuli was randomized, and this was repeated 20 times (20 repetitions per trial type).”

Motorized speaker:

Lines 664-665: “The stimulus sequence was randomized in a block design: all the positions were randomized, and this was repeated 20 times.”

A greater distance between successive stimuli might be more salient (and potentially induce saccades and larger corollary discharges). Is there any effect when the distance between successive stimuli is larger? (The same applies to both speaker array and movable speaker)

We analyzed the magnitude of the responses as a function of the distance to the previous stimulus (speaker or LED). This analysis showed that the distance of the previous stimulus does not influence the magnitude of the response. See reviewer Figure 1 below

Reviewer Figure 1. The population response magnitude did not depend on the distance between current speaker position and that of the previous stimulus. Gray, individual mice; black, average across mice; One-way repeated measure ANOVA: distance to previous speaker, $F(11,77) = 0.56$, distance to previous LED, $F(11,77) = 0.61$; $n = 8$ mice.

10) Fig. 1f shows that, while a large part of the connections is sending projections from AC and V2L to layer 1 (L1) of the visual cortex, the targets in deeper layers of the visual cortex are also

clearly evident by the fluorescent data (figure 1f). The authors should perhaps better discuss if it is possible that different results might have been obtained by focusing on projections to different V1 layers (if e.g. these came from different sources in A1). This focus on I1 could also be made evident in the title or abstract.

We change the abstract and the discussion accordingly.

Abstract (Lines 16-17): "Using two-photon axonal calcium imaging and a speaker array we measured the auditory spatial information transmitted from AC to layer 1 of V1."

Lines 410-412: "As our measurements were obtained from inputs terminating in L1, we cannot rule out the possibility that afferents terminating in deep layers (Figure 1e,f) might harbor different sound localization signals than those terminating in L1 or that they would be topographically organized."

11) Viral AC injections: It is unclear how this second injection site (3.0 mm, 4.7 mm 0.5 mm depth) would hit the auditory cortex. Why did you choose this dual injection strategy?

We aimed at covering a large fraction of AC. As AC is quite elongated along the AP axis and required two injections. The injection site at 3.0 mm, 4.7 mm 0.5 mm depth was selected to target the posterior AC cortex based on the Allen Atlas. Notice also that we quantitatively verified the precision of our injection in Supp Figure 1 and almost all GCaMP6 expression was confined within AC, as intended.

12) "Sound stimulation consisted of five 200 ms bursts with 10 ms raised-cosine onset offset ramps (1s total)". This is not entirely clear - what's the difference with a constant 1 s white noise burst?

We clarified this in methods. We also present a cartoon of the stimulus in Fig1c.

Lines 613-615: "Sound stimulation consisted of a continuous sequence of five bouts, each lasting 200 ms, with 10 ms raised-cosine onset/offset ramps ensuring smooth transitions. The entire sequence spanned a total duration of 1 s without any gaps between the bouts."

13) "For the analysis of the magnitude of the audiovisual modulation, we excluded neurons with high variability in their response (s.d of the response > 10, <1 % of all neurons)." Variability to a particular stimulus, or variability over the entire session? What is the reason for this exclusion?

We revised the method for selecting neurons and now use a more standard method. Neurons with fluorescence intensities that were very close to the neighboring neuropil value were removed. This is standard practice in the field as their weak fluorescence leads to a poor signal-to-noise ratio.

Line 876-877: "Only neurons with mean trial baseline fluorescence signal stronger than the surrounding neuropil signal by more than 3% were kept⁸⁹ (79.2±6.8%, mean ± sem, 8 mice)."

Reviewer #2 (Remarks to the Author):

It has been noted for a long time that auditory stimuli elicit responses in the mouse primary visual cortex (V1). However, the nature of the auditory information conveyed to V1, its organization, and its role in visual processing remain unclear. Recent evidence has shown that part of the response of V1 neurons to auditory stimuli is caused by uninstructed behavioral responses rather than the auditory stimuli themselves.

Projections from the primary auditory cortex (AC) are known to reach V1. This study aimed to investigate whether sound location information is present in AC projections to V1 and whether it is topographically organized, similar to the superior colliculus. Dual-color two-photon microscopy was employed to image calcium responses of AC boutons (and V2L boutons) in V1 and relate them to the responses of V1 neurons. The study found that AC boutons in V1 were often modulated by sound location, but the spatial extent of their responses varied greatly. As a control, the activity of V2L boutons in V1 was also imaged, revealing more confined visually induced responses, while auditory responses in these boutons were not spatially confined.

Contrary to V2L, AC boutons did not exhibit a topographic organization matching visual space in V1. However, sound location could be decoded based on AC bouton responses, albeit with better accuracy for azimuthal space than for elevation. In contrast, location decoding was not possible based on V2L bouton activity in response to sounds. The authors did not find evidence linking AC location responses to uninstructed behavior of the mice. Although AC activity often correlated with pupil size and facial movements, there was no relationship between facial movements and sound location, nor could sound location be decoded from V2L boutons. Furthermore, the study assessed whether the responses of V1 neurons were more strongly

modulated by sounds when the location of the sound and visual stimulus (or their brightness and loudness) were congruent, but no evidence supporting this hypothesis was found.

Taken together, this is a thoroughly executed study providing strong evidence for location-specific auditory input to V1 for the first time. The statistical analyses conducted were appropriate. However, it remains unknown how location-specific auditory information is actually used in V1 and what its behavioral relevance is.

We thank the reviewer's valuable insights and constructive recommendations.

Several issues still need attention.

Major issues:

1. In the Methods section, it is mentioned: "We recorded videos of the mouse eye/face during two-photon imaging." Please specify which data was available/analyzed for each group since it

appears that in some sets of animals, only the eyes were imaged, while in another set of animals (the CBA mice only?), full-face movies were recorded.

We added this information to the method section:

Lines 707-715: "The eye was recorded in experiments measuring visual and auditory 2D spatial RFs using the speaker/LED array with bandlimited white noise (5/8 AC- and all three V2L-injected mice). No face recordings were obtained from these mice. The face was recorded in experiments measuring auditory 2D spatial RFs using broadband auditory white noise (2/2 Thy1-jRGECO1a and 6/6 CBA mice). For the azimuth and loudness tuning experiments, eye recordings were obtained from 2/3 mice and face recordings from 1/3 mice. For the motorized speaker experiments, eye recordings were obtained from 2/5 mice and face recordings from 3/5 mice. When measuring audiovisual modulations in V1 neurons, we obtained eye recordings from 4/8 mice and face recordings from 4/8 mice."

2. To strengthen the claim that location-modulated responses of AC boutons are not related to uninstructed behavior of the mice, it would be useful to show that location-specific responses are also present in trials without facial movements.

We measured AC responses in sessions in which there was no significant face motion when stimulated with 2-80kHz noise. AC axons still showed location-specific responses and the location of the speaker was decoded from their activity.

We now show these analyses in Supplementary Figure 4g-j.

3. It is unclear whether the modulation of V1 responses by AC is directly caused by auditory inputs to V1 or indirectly by uninstructed behavior. To address this, it would be valuable to determine if the same result is obtained when only trials without facial movements are used or when facial movements are regressed out (if this data is available, please refer to point 1).

We found that, while facial movements were undetectable using the lowest sound levels, AV modulations persisted in those trials and were of the same amplitude as when using louder sounds. This finding suggests that unstructured movements alone are not the exclusive factor driving AV modulations in V1 neurons in our recordings. We now show these analyses in Supplementary Figure 10.

Nevertheless, we now acknowledge in the results that we do not know to what extent AC projections to V1 are involved in these AV modulations. We now directly address this in the results section:

Lines 319-332: "Sound elicited unstructured movements in a loudness-dependent manner. Loud sounds evoked reliable movements while quieter ones did not (Supplementary Figure 10e,f,h). In contrast to this, the magnitude of the AV modulation was similar across loudness levels (Supplementary Figure 10d). V1 neurons showed AV modulations and A-only responses at the lowest sound intensity even though sound-evoked movements were undetectable. Auditory

responses and AV modulation remained independent of the distance between the speaker position and the neuron's RF or visual stimulus, respectively, when using quiet sounds (Supplementary Figure 10g,i). This suggests that some components of the auditory-evoked activity and the sound modulations of the visual responses in V1 are motor-independent, as suggested previously 14,29. Whether these motor-independent signals depend on AC→V1 inputs remains to be determined. While we cannot completely dismiss the possibility that certain components of the signals may contain information regarding the spatial congruency of audio-visual stimuli or the positioning of sounds relative to the receptive fields of V1 neurons, these signals do not exert a predominant influence on V1 responses in mice when they are passively exposed to sounds."

See also the reply to a similar point raised by reviewer 1 .

4. In line 232, the authors state that Thy1-jRGECO1a mice were on a C57bl/6 background. Please provide more information about how the mouse line was maintained.

We now clarify this in the methods.

Lines 489-491: "Thy1-jRGECO1a mice were originally created on a C57BL/6J genetic background ⁴⁵ and the line was maintained in the same background by backcrossing heterozygous mice."

5. It would be useful to provide information regarding whether location-specific suppression of spontaneous activity was observed in the dark (considering Deneux et al., eLife 2019). Unfortunately, we can not perform these analyses as the recordings of AV modulation of V1 neurons were done under dim background illumination and not in the dark as in Deneux et al.

Minor issues:

1. There are several typos in the text. Please check lines 90, 198, 204, 310, 322, 329, 330, 343, 344, 384, 609, 623, 733.

We fixed all these typos.

2. The authors refer to Supplementary Figure 5, but it should be Supplementary Figure 4.

We fixed this mistake.

3. The sentence on line 310 is a bit confusing due to the mention of the lack of selectivity: "As in the AC as a whole, AC afferents were spatially non-selective, or selective to varying degrees to sounds in specific psi- or contralateral locations."

We corrected the sentence as suggested.

4. The authors sometimes refer to responses of boutons and sometimes of axons, which does not seem consistent. Please review and fix where necessary.

We identified and grouped boutons of the same axon using correlation analyses when decoding sound location. This is required as the Bayesian decoder assumes independence across units. Otherwise, we describe observations across boutons. We now clarify this in the results and methods sections and make sure we use the two words consistently with this.

Lines 156-159: "Within each individual imaging session, we grouped highly correlated boutons into one functional unit as they are likely to belong to the same axon^{49,50}. We subsequently refer to these functional units as axons."

Lines 775-776: "Activity in each cluster was assigned to the ROI with the largest mean $\Delta F/F$. We refer to these clusters as "axons"."

5. More information and analysis of onset and offset responses separately would be helpful. Are onset responses dominant, as suggested by Deneux et al.?

We now identified onset and offset responses. We found onset responses to be dominant and that sound location would be better decoded from onset than offset responses. We show this new analysis in Supplementary Figure 3.

6. The stimulus parameters are confusing (e.g., how can a 1-second stimulus consist of 5 200 ms bursts?). Please provide clearer descriptions.

We now clarified this in the methods.

Lines 615-617: "Sound stimulation consisted of a continuous sequence of five bouts, each lasting 200 ms, with 10 ms raised-cosine onset/offset ramps ensuring smooth transitions. The entire sequence spanned a total duration of 1 s without any gaps between the bouts."

7. In line 111, the authors state: "We confirmed that AC boutons were frequency-tuned, as expected from auditory responses in AC neurons (Supplementary Figure 2a)." However, only three examples are shown, and no quantification is provided to support this statement.

We expanded our analyses in Supplementary Figure 2 and now show population analyses on the fraction of frequency-tuned AC boutons as well as the distribution of best frequencies.

8. Please ensure that the statistics used are always clear and complete. Statements like "The position eliciting the largest response varied across AC axons, with a slight enhancement in boutons tuned to high-azimuth positions (Fig. 2g)" (line 152) need to be supported by statistics.

We fixed this mistake. We now provide supporting statistics in Figure 5 for the larger proportion of boutons tuned to high ipsi- and contralateral azimuthal positions when describing the results.

Lines 276-282: "Consistent with the measurements using the speaker array, many AC→V1 axons were spatially tuned, and neighboring boutons showed different spatial RFs that, in many cases, peaked in different hemifields (Figure 5b). Boutons tuned to high (>30°) contralateral and ipsilateral locations were more abundant than those tuned to frontal locations (Figure 5c). Consequently, the locations of sounds originating in lateral locations in both hemifields could be decoded with similar error, but those in front of the animals resulted in larger decoding errors (Figure 5d)."

9. In line 189: "First, these modulations are low dimensional, i.e., they affect all neurons similarly, while the spatial profile of the sound-evoked activity in AC to V1 inputs was diverse across boutons, even within the same session (Figure 2a)." It seems to be referring to the wrong figure or panel.

We fixed this mistake.

10. Please adjust the scales in Supplementary Figure 4b and 4c so that the data can be assessed more easily.

We adjusted the scales.

Reviewer #3 (Remarks to the Author):

In this carefully executed study, Mazo et al. have managed to record simultaneously in mouse primary visual cortex the inputs coming from auditory cortex and the activity of V1 neurons while playing sounds and visual stimuli arriving from different spatial locations. Thanks to these experiments they could show that spatial information about the sound reaches visual cortex but is not spatially aligned with retinotopic information. This result is important, as it indicates that alignment of visual and auditory spatial receptive fields, at least in mice, is specific to the superior colliculus. It is of course not possible to exclude that it is a mouse-specific result, potentially related to the poor accuracy of spatial hearing in mice, however this result is extremely useful for understanding the mechanisms underlying in orienting behaviors based on multisensory cues.

I recommend nevertheless a few verifications in order to make sure that weak RF alignment has not escaped the careful analysis of the data and to evaluate the generality of this results.

We thank the reviewer for the useful, constructive comments and suggestions.

1/ If I understood well, the authors have evaluated the spatial alignment of receptive fields with two methods. The direct one compares the averaged best azimuth of a AC axonal population against the average best azimuth for visual stimuli for the underlying V1 neurons. This method is likely to be highly sensitive to response variability, as the best azimuth of a given axon is not measured with high certainty when the axon itself is not sharply tuned. Particularly, the

contribution of weakly tuned axons may increase variability and hide local population tuning. To verify that it is not the case, I recommend to perform the analysis in the opposite order. First average responses across all axons and then compute the preferred azimuth for comparison with V1 neurons. This population measure would certainly complement the classifier analysis used by the authors, which is supposed to address the point, but may do this less accurately and directly than the method I propose, due to the sensitivity of classifiers to noise in high dimensions.

We conducted an analysis to measure the spatial alignment between the population auditory receptive field and the visual receptive field of V1 neurons, as recommended. We averaged the responses across all boutons and subsequently calculated the best azimuth from this data. Similar to our previous method, we found that the auditory receptive fields of AC axons lack topographical alignment and show no correlation with the receptive fields of V1 neurons.

We included this analysis in Supplementary Figure 7

Lines 219-220: "A similar result was obtained when we first averaged responses across boutons to compute the population best azimuth (Supplementary Figure 7)."

2/ Earlier studies have suggested that auditory inputs to V1 depend strongly on intensity. Here the intensity used 66dB for C57 mice and ~50 dB for CBA mice is rather in the low range. If the authors have an indication whether the lack of spatial RF alignment is still seen at higher intensity (80dB in C57 or 70 dB in CBAs), it would significantly strengthen their results.

With our current speakers and amplifiers, it is not feasible to generate white noise within the 2-20 kHz frequency range with a flat frequency power spectrum exceeding 85 dB at the source (or 65 dB when using higher frequencies). Nevertheless, 85 dB at the source, which results in approximately 66 dB at the mouse's ear, appears to align with the range employed in prior research. For instance, Deneux et al. employed sound levels ranging from 50 to 85 dB at the ear, while Knöpfel et al., Iurilli et al., and Ibrahim et al. utilized sound intensities of 70-72 dB (though it remains unspecified in these papers whether this measurement was taken at the source or the ear location). These values fall within the range of sound intensity observed at the ear in our measurements. Consistently, we were able to observe robust auditory evoked responses at this sound intensity.

While we cannot stimulate at higher intensities with our current setup, we conducted new measurements of the spatial receptive fields and retinotopic alignment of AC→V1 inputs at various decreasing sound intensity levels. We found that at lower sound intensities AC boutons still showed spatially specific sound evoked responses that a decoder could use for sound localization. These location-specific sound-evoked responses remained non-topographical organization when using lower intensities.

We now show these new measurements in Supplementary Figure 5.

We also modified the results sections accordingly:

Lines 168-169: "Spatially specific responses were also evoked in AC boutons using lower sound levels, and speaker location could be decoded from them (Supplementary Figure 5-a-d)."

Lines 219-221: "A similar result was obtained when we first averaged responses across boutons to compute the population best azimuth (Supplementary Figure 7) and when we measured best azimuth using lower sound levels (Supplementary Figure 5e)".

3/ Line 351 in the discussion, the authors state that V1 does not send inputs to AC. This statement should be mitigated. Connectivity studies show clearly an asymmetry but do not fully exclude weak inputs from V1 to AC. Functional studies show clear visual responses in deep layers of AC, e.g. the recent work of the Hasenstaub lab.

We have revised this paragraph to provide a clearer statement that V1 indeed innervates AC, although the inputs in this direction are comparatively less abundant than in the reverse direction.

Lines 404-406: "On the other hand, AC inputs to V1 are weakly reciprocated, unlike those from higher-order visual and frontal areas, as AC inputs to V1 are considerably stronger than those traveling in the opposite direction in mice and other species^{28,67}."

4/ Two suggestions to be included in the discussion. As the author mentioned a recent study show that spatial information about sounds is more efficiently extracted with high frequency information. The Kanold lab has demonstrated few years ago that despite have much better high frequency hearing CBAxC57 mice still have weak high frequency representations in cortex. <https://www.nature.com/articles/s41598-020-67819-4> It may explain why the auditory visual spatial mapping does not emerge in mice at the cortical level.

To the best of our knowledge, the study mentioned reports that CBAxC57 mice have more neurons representing high frequencies in AC than C57 mice. We now mention this when introducing the experiments with CBA mice.

It may be that in the lab mouse usual sound scape (small cage likely with reverberations), the directionality information carried by lower frequency is quite unreliable, preventing alignment, while the more local high frequency are more reliable.

Another idea in the same direction, given the poor accuracy of sound localization in mice it may not be advantageous to add this information to the already accurate visual representation of events in space.

We now address these points in the discussion:

Lines 441-444: “In spatial tasks, the visual sense often prevails as it harbors more precise spatial information ⁷⁵. Hence, the lack of topographic organization in AC→V1 inputs might reflect that, while integrating sound information with visual inputs, V1 is not engaged in the detection of low-level features, such as the location of sounds.”

Lines 449-451: “We cannot dismiss the possibility that topographic AC→V1 projections might develop in mice exposed to a more diverse multisensory environment than what is typically found in standard laboratory home cages.”

Reviewer #4 (Remarks to the Author):

Although projections from auditory to primary visual (V1) cortex are well established, the role of these projections remains a significant gap in our understanding of audiovisual integration. Here, Mazo et al convincingly test the hypothesis that auditory projections to V1 are topographically matched, and establish that this is not the case: although axons from auditory cortex did demonstrate spatial selectivity, this was not correlated with the receptive field of the colocalised population of V1 neurons. To demonstrate that their labelling technique and analyses could identify sensory-map alignment if it was present, the authors reproduced the result that spatial responses of V2L-axons do correspond to the spatial selectivity of the colocalised V1 population. The authors further demonstrate that the signals from auditory cortex are only weakly lateralised, and that spatial coherence between auditory and visual stimuli does not significantly impact the auditory modulation of V1 activity. The authors convincingly deal with concerns regarding audio-evoked movement signals.

These data represent an important, and original, contribution to the field. Aside from the issues (one major, several minor) listed below, I found the methodology and analyses convincing, and believe the central thesis of the paper is supported by the data.

We thank the reviewer for the constructive comments and suggestions.

Major concerns:

1) In several experiments (e.g. Fig. 4b), the authors claim to play sounds of “2-80 kHz auditory white noise” with their speakers (MULTICOMP PRO, # ABS-239-RC). However, these speakers are only rated to deliver sounds up to 20 KHz. Typically speakers that can reliably produce frequencies at the top end of this range cost hundreds of euros. Therefore, speaker calibration plots, including the accurate reproduction of sinusoids in this high-frequency range, need to be provided as supplemental data to demonstrate that these speakers are capable of delivering the sounds that the authors claim to produce.

The speakers employed have a specified frequency rating of up to 20 kHz; however, they are indeed capable of reproducing frequencies up to 80 kHz, albeit with some harmonic distortions. (Reviewer Figure 2).

Note that the only experiments conducted using pure tones are in Supplementary Figure 2, where we characterized the frequency tuning of AC boutons in V1. However, we used pure tones with frequencies < 20 kHz. Reviewer Figure 2a,b shows the example responses of the speakers to 20 kHz and 80 kHz pure tones, showing that they were able to generate pure tones up to 20 kHz and the harmonic distortions produced at 80 kHz. Despite the harmonic distortions at high frequencies (>20 KHz), the speakers were capable of producing 2-80 kHz white noise after calibration. The white noise stimulus, with a flat spectral response up to 80 kHz, obtained from each of the 39 speakers after an example round of calibration is now shown in Supplementary Figure 11.

We also now more extensively discuss in Methods that our setup can produce white noise 2-80 kHz white noise, but is not able to reproduce high-frequency pure tones without introducing distortion given the limitation of the speakers:

Lines 587-590: “Notice that, while the frequency response to white noise up to 80 kHz was flat after calibration (Supplementary Figure 11), the speakers in the array are specified for frequencies up to 20 kHz. Consequently, they cannot produce pure tones of frequencies higher than 20 kHz without introducing harmonic distortions.”

Reviewer Figure 2. a-b 20 kHz tone. **a** Waveform recorded (blue) vs. sin wave generated at the same sample frequency (red, 192 kHz). **b** Spectrogram of the recorded 20 kHz tone. **c,d** same for 80 kHz tone

Minor concerns:

1) Although I do believe the result that projections from auditory cortex are spatially selective, it would strengthen this finding to show that it was not specific to the type of auditory stimulus presented (e.g. showing that within a session, the spatial selectivity of a neuron is not highly dependent on the frequency spectra). This would reassure the reader that differences in response to not reflect differences in the speakers themselves which may remain even after calibration.

We expect spatial selectivity to be dependent on the frequency content of the sound, given that sound localization in mice depends on high-frequencies (e.g. Allen and Iso 2010).

Thus, to confirm that the spatial selectivity does not reflect differences in the sound produced across speakers, we repeated the measurements with a single speaker that rotates around the head of the mice. These experiments confirmed both our findings that AC→V1 inputs are spatially selective and that they are not topographically organized in V1, independently of any possible spectral differences across speakers.

Another observation that further confirms that the difference in responses across speakers is due to spatial tuning is that the 2nd best azimuth response is next to the best speaker/location (Supplementary Figure 2e). If responses were due to differences in the speakers themselves, it wouldn't be expected as the best and second best azimuthal responses wouldn't be necessarily adjacent to each other.

We now modified the text to make these points more clear:

Line 257-268: **"Spatially specific sound responses in AC→V1 inputs are not due to differences across speakers.**

We wondered if, despite having calibrated the frequency responses across speakers, the spatially specific sound responses of AC axons could still reflect unnoticed variations in the sounds produced by the different speakers in the array. To ensure that this was not the case, we mounted a single loudspeaker on a rotating arm and presented sounds at different azimuthal positions (Supplementary Figure 9a). We confirmed that the spatially specific sound responses of AC axons obtained with the speaker array were similar to those measured using the rotating speaker, as assessed by their SMI and the performance in decoding azimuthal position (Supplementary Figure 9b,c). Furthermore, like when using the speaker array for stimulation, the spatial auditory responses in AC→V1 inputs lacked topographical organization when evaluated using a rotating speaker (Supplementary Figure 9d)."

2) First sentence of abstract: "Perception requires binding spatiotemporally congruent multimodal sensory stimuli." One can perceive the world without binding stimuli. The first sentence of the intro is also too general (many animals don't have multiple specialised sense organs).

We modified each of the sentences to reduce their generality.

3) In figure legends (e.g. Fig 2d) it reduces readability to report all p values exactly and state the comparisons. When highly significant, I would suggest stating e.g. *** = $p < 0.001$ at the end of the figure.

We did our best to improve the readability of the figure legends. However, we need to keep exact p values, as it is required in the journal's guidelines.

4) Fig 3b why is a threshold used instead of simply showing the distance of the estimate from the sound location?

We now show the decoding error in all figures, as suggested.

5) Related to the above, in Supplemental Figure 3a, I am concerned about the accuracy of the shuffle. The "chance level" decreases at -20/100 degrees (presumably because there are less "correct" guesses at these locations because of the thresholding). However, why does the shuffle accuracy show no corresponding decrease? Also, why is the shuffle more accurate than chance for all locations in the V2L axons of this figure?

The reviewer is correct, the chance level decreases at -20/100 degrees because there are fewer correct guesses.

The shuffle was done across axon identities, not across trials. As such, shuffle accuracy reflects reflected population biases in the tuning and not the chance level. For the same reason, the decoder using V2L visual responses performs higher than chance for the LED at the position that is retinotopically matched with V1 RF, as V2L visual responses are retinotopically matched on average.

We agree that this was confusing. In the revised MS we now shuffle the data across trials. The decoding error using the shuffled data now reflects the chance level in Figures 3, 4, and 6, and Supplementary Figures 4a, 6, and 8c, as expected.

6) Related to the above, in Supp. Fig 5c, why are the chance levels different from 3a?

In the previous version, the chance levels appeared to be different when they were not as the two plots were plotted with a different Y-axis range in Supp. Fig 5c and 3b (bottom). We changed the plots in those panels from the previous version as explained in the previous point (shuffling is now done across trials, not axons as in the previous version). We now plot Supplementary Figure 8c in the same way as in Figure 5c.

7) "We scored the accuracy of the decoder by measuring the fraction of times that the decoded position with the largest likelihood was within 10° of the actual one." This is only true to azimuth--not elevation--I believe? But this isn't stated in the main text.

Correct, decoding had to be on the actual elevation position to be considered accurate in elevation. These plots have now been updated to show the distance between the estimated to actual sound location as suggested by the reviewer above.

8) “with a only slight , but significant, overrepresentation of contralaterally-tuned ones (Figure 5c,d).” Unless I missed it, there is no mention of this significance, or the significance test, in the figure?

We modified this statement. In the revised version we increased the number of mice analyzed in Figure 5. We now show that boutons tuned to frontal areas are less abundant than those tuned to large azimuth contra- and ipsilateral locations and show decoding analysis with its associated significance test in the Figure.

9) Sup. Fig. 2a: does the changing (white to black) on the frequency bar at the bottom mean anything? If not, I would remove this bar completely, especially as a thickening bar is more associated with loudness/intensity than frequency.

We removed the thickening bar in Supplemental Figure 2 as suggested.

10) Missing references:

a. [https://www.cell.com/neuron/fulltext/S0896-6273\(17\)30007-7](https://www.cell.com/neuron/fulltext/S0896-6273(17)30007-7) where activation of AC > V1 projections did not result in suppression of V1 (in contrast to Ref 15 in the ms).

b. <https://pubmed.ncbi.nlm.nih.gov/37295419/> where visual cortex inactivation doesn't impact auditory spatial localization in mice.

c. <https://pubmed.ncbi.nlm.nih.gov/32402272/> where higher visual regions are suggested to more closely reflect audiovisual detection behaviour.

We now cite and discuss these references.

11) There are a lot typos in the manuscript. I have noted several here, but have likely missed others. It warrants a careful read before publication:

a. Line 35 “stablished”

b. Fig2a: should not be “across” individual mice.

c. Line 679 “was” should be “were”

d. Line 204 “their spatially pattern in ACàV1 boutons”

e. Line 224 refers to “LM” for the only (I think) time in the paper

f. Line 262 “with a only slight”

g. Line 311 “specific psi- or contralateral locations”

h. Line 325 “a large fraction of AC inputs was”

i. Line 326 “for ipsilaterally sounds originating”

j. Line 330 “(2-80 kHz white noise”—no closing bracket

- k. Line 333 “with an accurate spatial map as V1”
- l. Line 337 “enhanced by spatially congruent bimodal stimulus”
- m. Capitalization of axes labels are inconsistent throughout the ms/supp

We fixed all these typos.

REVIEWERS' COMMENTS

Reviewer #1 (Remarks to the Author):

The authors addressed all my comments satisfactorily. I believe that the manuscript has now significantly improved and that all confounding factors have been addressed or discussed.

Reviewer #2 (Remarks to the Author):

The authors have addressed all issues raised and I have no more questions. The paper is acceptable for publication in Nature Communications in my opinion.

Reviewer #3 (Remarks to the Author):

The authors have addressed appropriately all my comments.

Reviewer #4 (Remarks to the Author):

The authors have done an admirable job of addressing my concerns with the manuscript (and those of the other reviewers): both in terms of the framing/language and wrt methods and statistical analyses. I think the manuscript has significantly improved in clarity--particularly Fig 3 and the related supplements. The paper represents an important contribution to the field and the continued efforts to understand the role of A1>V1 connections. I have no further questions.